# Understanding Sampler Stochasticity in Training Diffusion Models for RLHF

## Abstract

Reinforcement Learning from Human Feedback (RLHF) is increasingly used to fine-tune diffusion models, but a key challenge arises from the mismatch between stochastic samplers used during training and deterministic samplers used during inference. In practice, models are fine-tuned using stochastic SDE samplers to encourage exploration, while inference typically relies on deterministic ODE samplers for efficiency and stability. This discrepancy induces a **reward gap**, raising concerns about whether high-quality outputs can be expected during inference. In this paper, we theoretically characterize this reward gap and provide non-vacuous bounds for general diffusion models, along with sharper convergence rates for Variance Exploding (VE) and Variance Preserving (VP) Gaussian models. Methodologically, we adopt the generalized denoising diffusion implicit models (gDDIM) framework to support arbitrarily high levels of stochasticity, preserving data marginals throughout. Empirically, our findings through large-scale experiments on text-to-image models using denoising diffusion policy optimization (DDPO) and mixed group relative policy optimization (MixGRPO) validate that reward gaps consistently narrow over training, and ODE sampling quality improves when models are updated using higher-stochasticity SDE training.

## 1 Introduction

Diffusion models (e.g., Stable Diffusion (Rombach et al., 2022), SDXL (Podell et al., 2024), FLUX (Black Forest Labs, 2024)) have shown strong performance in text-to-image (T2I) tasks, and have also been extended beyond images to video (Ho et al., 2022) and audio (Liu et al., 2023). To meet downstream objectives such as aesthetics, safety, and alignment, it is essential to post-train with RLHF (Ouyang et al., 2022) for preference-driven improvements, often with a KL-regularization term to preserve performance on pretrained tasks (Schulman et al., 2017). Widely used RLHF algorithms include DDPO (Black et al., 2024) and GRPO (Shao et al., 2024) variants (FlowGRPO (Liu et al., 2025), DanceGRPO (Xue et al., 2025), MixGRPO (Li et al., 2025)). DDPO directly optimizes human-preference rewards by casting the denoising process as a Markov Decision Process (MDP); GRPO variants use group-relative advantages. RLHF training may also exhibit unstable trajectories, long inference times, and vulnerability to reward hacking (Skalse et al., 2022; Lee et al., 2025) and efficient, robust samplers (Lu et al., 2022; 2025) are desired for stable, high-quality fine-tuned models. See (Winata et al., 2025) for a broader review of successful RLHF algorithms for generative models.

The classical scheme DDPM (Ho et al., 2020), a discretization of the score-based backward SDE (Risken, 1996; Song et al., 2021b), allows for trajectory diversity and rich exploration in RLHF training; the deterministic DDIM sampler (Song et al., 2021a) follows a probability-flow ODE and facilitates inference. High stochasticity during training is crucial for effective exploration of the reward landscape. However, during inference, deterministic ODE sampling is preferred to ensure consistency and computational efficiency. This creates an inherent tension: we train via a highly stochastic process (SDE) but we deploy a deterministic process (ODE) for inference. This discrepancy necessitates the following question:

> **To what extent can we guarantee ODE inference quality after we fine-tune diffusion models with SDE sampled trajectories?**

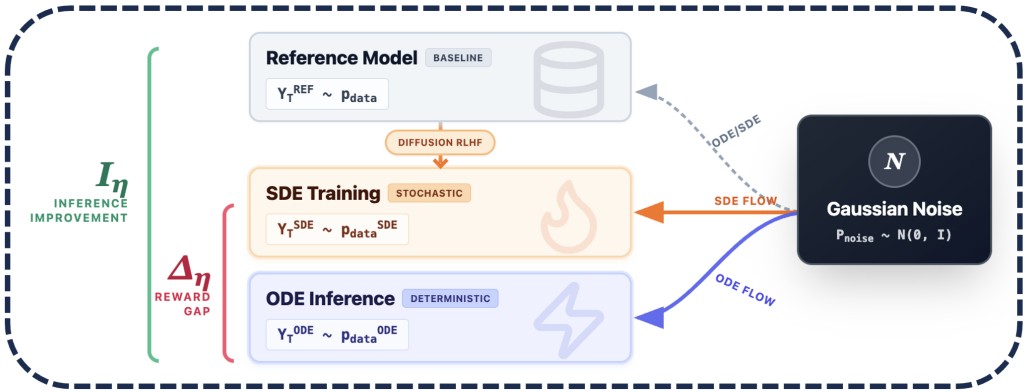

**Figure 1:** An illustration of the problem formulation. $\Delta_\eta$ represents the "**Reward Gap**" and $I_\eta$ represents the ODE reward improvement after SDE fine-tune. We would like $\Delta_\eta \downarrow$ with $I_\eta \uparrow$.

Figure 1 illustrates the problem formulation. While SDE and ODE denoising processes (equation 3) yield the same terminal marginal for the pretrained model (see Section 2.1), a *distributional difference* arises between the distribution ($p_{\text{data}}^{\text{SDE}}$) *after* stochastic fine-tuning and the deterministic inference outcome ($p_{\text{data}}^{\text{ODE}}$) (see Section 3). We denote $\eta$ as the stochasticity scale, where $\eta = 1.0$ corresponds to the standard variance in DDPM. We define this discrepancy as the **Reward Gap** ($\Delta_\eta$), which must be bounded to preserve the ODE reward improvement ($I_\eta$) gained over the baseline. In this paper, we present our analysis from three perspectives:

- **Theory**: We bound $\Delta_\eta$ between a generally SDE-fine-tuned model and its ODE-sampling counterpart using Gronwall's inequality. Specifically, for VE, VP and mixture Gaussian models, the gap shrinks at sharp rates $O(1/T)$ and $O(e^{-T^2}/2)$, where $T$ is the denoising time horizon.

- **Methodology**: To handle high stochasticity beyond $\eta = 1.0$ without changing marginals, we adopt the gDDIM scheme (Zhang et al., 2023) for arbitrary stochasticity levels. This framework is introduced in Section 2.2 and 5. See Appendix E for more technical details.

- **Experiments**: We evaluate large-scale T2I models and RLHF algorithms, DDPO and Mix-GRPO, to quantify the reward gap across multiple reward functions. The results show that $\Delta_\eta$ decrease as training quality improves and $\eta = 1.2$ yields superior ODE inference performance, $I_\eta$.

**Related Literature.** Higher-order ODE solvers such as RX-DPM (Choi et al., 2025), DEIS (Zhang & Chen, 2022) approximate probability-flow. Recent analyses give broader convergence guarantees (Huang et al., 2025) and better representability (Chen et al., 2023b) for score-based diffusion. In parallel, RL-based fine-tuning leverages stochastic sampling: DRaFT differentiates through noisy trajectories (Clark et al., 2023), Score-as-Action casts fine-tuning as stochastic control (Zhao et al., 2025), and Adjoint Matching enforces optimal memoryless noise schedules (Domingo-Enrich et al., 2024). Large-scale studies combine multiple rewards (Zhang et al., 2024a); SEPO, D3PO, and ImageReFL boosts alignment (Zekri & Boullé, 2025; Yang et al., 2024; Sorokin et al., 2025). Finally, (Liang et al., 2025) gives discretization error bound and (Wu et al., 2024) discusses guidance (Ho & Salimans, 2021) for Gaussian Mixture priors.

**Organization of the paper.** The remainder of the paper is organized as follows. In Section 2, we provide background on diffusion RLHF and gDDIM in particular. Section 3 computes *exact* $\Delta_\eta$ and $I_\eta$ for simple controls. More complex networks are analyzed in Section 4. Numerical experiments in Section 5 validate our theoretical bounds by fine-tuning large-scale T2I models with DDPO and GRPO algorithms. Conclusion is given in Section 6.

## 2 PRELIMINARIES

### 2.1 FORWARD-BACKWARD DIFFUSION MODELS

The goal of diffusion models (Song et al., 2021b) is to generate new samples similar to $p_{\text{data}}(\cdot)$. The forward process is given by an SDE:

$$dX_t = f(t, X_t)dt + g(t)dB_t, \quad X_0 \sim p_{\text{data}}(\cdot), \tag{1}$$

where $\{B_t\}$ is the $d$-dimensional Brownian motion, and $f : \mathbb{R}_+ \times \mathbb{R}^d \to \mathbb{R}^d$ and $g : \mathbb{R}_+ \to \mathbb{R}_+$ are given model parameters under regularity conditions (Stroock & Varadhan, 1979).

We use *denoising score matching* (Vincent, 2011) to minimize the mean square error between a parametrized neural network $s_\theta(t, X_t)$ and the unknown term $\nabla \log p(T - t, \overline{X}_t)$ in the time reversal SDE (Haussmann & Pardoux, 1986). The reverse-time SDE becomes:

$$dY_t = \left( -f(T - t, Y_t) + g^2(T - t)s_\theta(T - t, Y_t) \right) dt + g(T - t)dB_t, \quad Y_0 \sim p_{\text{noise}}(\cdot). \tag{2}$$

For richer levels of *sampler stochasticity*, a flexible stochasticity parameter $\eta \geq 0$ is introduced:

$$dY_t = \left( -f(T - t, Y_t) + \frac{1 + \eta^2}{2} g^2(T - t)s_\theta(T - t, Y_t) \right) dt + \eta \, g(T - t)dB_t, \quad Y_0 \sim p_{\text{noise}}(\cdot), \tag{3}$$

With a divergence-free noise scale, equation 3 preserves the terminal marginals (Anderson, 1982).

### 2.2 DISCRETIZATION – STOCHASTIC GDDIM

As shown in (Song et al., 2021b), for most diffusion models, the model parameters are:

$$f(t, x) = \frac{1}{2} \frac{d \log \alpha_t}{dt} x \quad \text{and} \quad g(t) = \sqrt{-\frac{d \log \alpha_t}{dt}},$$

where $\{\alpha_t\}_{t=0}^T$ is a decreasing sequence from 1 to 0. For all $\eta \geq 0$, (Zhang et al., 2023) proposed the *generalized* DDIM update:

$$x_{t-\Delta t} = \sqrt{\frac{\alpha_{t-\Delta t}}{\alpha_t}} x_t + \left( \sqrt{\frac{\alpha_{t-\Delta t}}{\alpha_t}} (1 - \alpha_t) - \sqrt{(1 - \alpha_{t-\Delta t} - \sigma_t^2)(1 - \alpha_t)} \right) s_\theta(t, x_t) \\ + \sigma_t(\eta) \mathcal{N}(0, I), \quad x_T \sim p_{\text{noise}}(\cdot). \tag{4}$$

where $\{x_t\}_{t=0}^T$ is the discretized sequence of backward diffusion $Y_t$ and

$$\sigma_t(\eta) = (1 - \alpha_{t-\Delta t}) \left( 1 - \left( \frac{1 - \alpha_{t-\Delta t}}{1 - \alpha_t} \right)^{\eta^2} \left( \frac{\alpha_t}{\alpha_{t-\Delta t}} \right)^{\eta^2} \right). \tag{5}$$

For any $\eta > 0$, the discrete scheme equation 4 has terminal marginal equal to equation 3. We use this discretization formulation for post-training with $\eta > 1.0$ in Section 5. Details in Appendix E.

### 2.3 DIFFUSION RLHF

RLHF was originally proposed for LLM alignment (Bai et al., 2022; Ouyang et al., 2022). (Black et al., 2024; Fan et al., 2023; Lee et al., 2023) first formulated the denoising step as Markov decision processes (MDPs) to fine-tune diffusions. (Gao et al., 2024; Zhao et al., 2024; 2025) developed an RL approach based on continuous-time models. Here we focus on *Denoising Diffusion Policy Optimization* (DDPO) and *Group Relative Policy Optimization* (GRPO).

**DDPO** (Black et al., 2024): We formulate the denoising steps $\{x_T, x_{T-1}, \cdots, x_0\}$ as an MDP:

$$s_t := (\boldsymbol{c}, t, x_t), \qquad \pi(a_t|s_t) := p_\theta(x_{t-1}|x_t, \boldsymbol{c}), \qquad \mathbb{P}(s_{t+1}|s_t, a_t) := (\delta_{\boldsymbol{c}}, \delta_{t-1}, \delta_{x_{t-1}}),$$

$$a_t := x_{t-1}, \qquad \rho_0(s_0) := (p(\boldsymbol{c}), \delta_T, \mathcal{N}(0, I)), \qquad R(s_t, a_t) := \begin{cases} r(x_0, \boldsymbol{c}) & \text{if } t = 0, \\ 0 & \text{otherwise,} \end{cases}$$

where $\delta_\bullet$ is the Dirac point mass concentrating at $\bullet$. The objective of DDPO is to maximize:

$$\mathcal{J}_{DDPO}(\theta) := \mathbb{E}_{\boldsymbol{c}\sim p(\boldsymbol{c}), x_0\sim p_\theta(x_0|\boldsymbol{c})}[r(x_0, \boldsymbol{c})]. \tag{6}$$

The policy gradient of equation 6 is $\nabla_\theta \mathcal{J}_{DDPO} = \mathbb{E}\left[\sum_{t=0}^T \nabla_\theta \log p_\theta(x_{t-1}|x_t, \boldsymbol{c})\, r(x_0, \boldsymbol{c})\right]$ (Williams, 1992). A common parameterization of $p_\theta$ is isotropic Gaussian (Mohamed et al., 2020).

**GRPO** (Shao et al., 2024): GRPO first computes the *advantages*:

$$A_i := \frac{r(x_0^i, \boldsymbol{c}) - \frac{1}{G}\sum_{k=1}^G r(x_0^k, \boldsymbol{c})}{\mathrm{std}\left(\{r(x_0^k, \boldsymbol{c})\}_{k=1}^G\right)},$$

where $G$ is the group size; $\mathrm{std}\left(\{r(x_0^k, \boldsymbol{c})\}_{k=1}^G\right)$ denotes the *standard deviation* of group rewards. The objective of GRPO is to maximize:

$$\mathcal{J}_{GRPO}(\theta) = \mathbb{E}\left[\frac{1}{G}\sum_{i=1}^G \frac{1}{N}\sum_{n=1}^N \min\left(\rho_t^i(\theta)A_i, \mathrm{clip}(\rho_t^i(\theta), 1-\epsilon, 1+\epsilon)A_i\right)\right], \tag{7}$$

where $\mathrm{clip}(\rho_t^i(\theta), 1-\epsilon, 1+\epsilon) := \min\{\max\{\rho_t^i(\theta), 1-\epsilon\}, 1+\epsilon\}$ restricts the likelihood ratio $\rho_t^i(\theta) = \frac{p_\theta(x_{t-1}^i|x_t^i, \boldsymbol{c})}{p_{\theta_{\mathrm{old}}}(x_{t-1}^i|x_t^i, \boldsymbol{c})}$ within the range $[1-\epsilon, 1+\epsilon]$ by imposing hard constraints on the boundaries; $\{x_t^i\}$ is the $i$-th sample trajectory in the group; $N$ is the number of grouped outputs, and the expectation is with respect to $\boldsymbol{c}\sim p(\boldsymbol{c})$ and $\{x_t^i\}\sim \pi_{\theta_{\mathrm{old}}}(\cdot|\boldsymbol{c})$.

## 3  THEORY ON THE REWARD GAP

As discussed in Section 1, forward–backward diffusion models typically employ *stochastic* SDE dynamics for exploration in the RLHF training, while adopting *deterministic* ODE samplers for inference (see Figure 1). In practice, we follow entropy-regularized objectives to fine-tune a parametrized family of score functions (Uehara et al., 2024; Tang, 2024):

**Definition 3.1.** *Define*

1. $\{Y_T^{\mathrm{REF}}\}$ *as the samples extracted from the referenced base model following equation 3 with terminal marginal $p_{data}$;*

2. *We fine-tune the following objectives under a fixed stochasticity parameter $\eta$:*

$$F_\eta(\theta) = \mathbb{E}[r(Y_T^\theta)] - \beta\, \mathrm{KL}(Y_T^\theta \,\|\, Y_T^{\mathrm{REF}}), \qquad \theta_\eta^* := \arg\max_\theta F_\eta(\theta); \tag{8}$$

*let $\{Y_t^{\mathrm{SDE}}\} = \{Y_t(\theta_\eta^*)\}$ be the optimal fine-tuned model, and $\{Y_t^{\mathrm{ODE}}\}$ be the deterministic sampler obtained by letting $\eta = 0$:*

$$dY_t^{SDE} = \left(-f(T-t, Y_t^{SDE}) + \frac{1+\eta^2}{2}g^2(T-t)s_{\theta_\eta^*}(T-t, Y_t^{SDE})\right)dt + \eta\, g(T-t)dB_t,$$

$$dY_t^{ODE} = \left(-f(T-t, Y_t^{ODE}) + \frac{1}{2}g^2(T-t)s_{\theta_\eta^*}(T-t, Y_t^{ODE})\right)dt. \tag{9}$$

Although a pretrained model admits identical terminal marginals for arbitrary $\eta$ in the form of equation 3, once the data distribution is aligned to a human-preference reward $r(x)$, the distribution of $\{Y_T^{\mathrm{ODE}}\}$ and $\{Y_T^{\mathrm{SDE}}\}$ (we denote as $p_{data}^{\mathrm{SDE}}$ and $p_{data}^{\mathrm{ODE}}$) are not necessarily the same.

Under a fixed noise level $\eta$, the terminal marginal satisfies a *reward-tilting* distribution $p_{data}^{\mathrm{SDE}} \propto p_{\mathrm{data}}(x)\exp(r(x)/\beta)$ (Zhang et al., 2024b). The terminal score function

$$s_{\theta_\eta^*}(T, x) \approx \nabla \log p_{\theta_\eta^*}(T, x) = \nabla \log p_{\mathrm{data}}(x) + \frac{\nabla r(x)}{\beta},$$

but the fine-tuned parameter $\theta_\eta^*$ does not regulate the intermediate trajectory, as the RLHF objective equation 8 only considers the terminal marginal. Therefore, we could not establish a general relationship between $s_{\theta_\eta^*}(t, x)$, $r(x)$, and $\nabla \log p_{\theta_\eta^*}(t, x)$ for $0 < t < T$. In general,

$$\nabla \cdot \left(p_{\theta_\eta^*}(t, x) \cdot s_{\theta_\eta^*}(t, x)\right) \neq \Delta p_{\theta_\eta^*}(t, x).$$

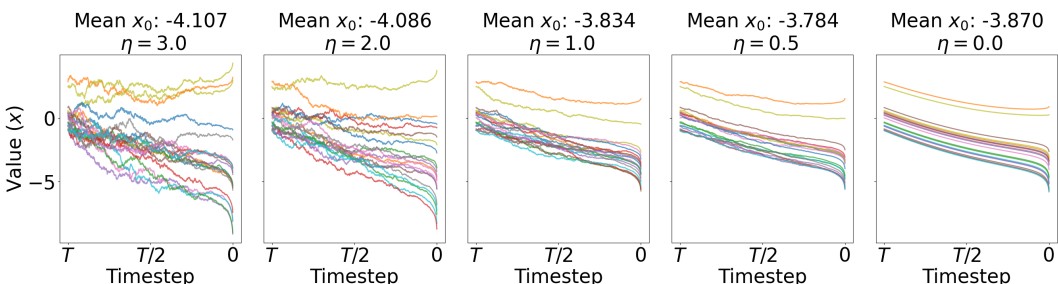

**Figure 2:** One-dimensional VP dynamics under different noise levels $\eta$, using the same control function, demonstrating $\mathbb{E}(Y_T^{\text{SDE}}) \approx \mathbb{E}(Y_T^{\text{ODE}})$ despite $p_{data}^{\text{SDE}} \neq p_{data}^{\text{ODE}}$.

As the reverse-time Fokker–Planck correspondence no longer satisfies the divergence free condition (Anderson, 1982), $p_{data}^{\text{SDE}}$ and $p_{data}^{\text{ODE}}$ need not coincide.

However, we are motivated by an observation in Figure 2 that 1D Variance Preserving (VP) dynamics, when run with the same constant downward drift, maintain a similar mean value regardless of the noise level $\eta$. we next rigorously quantify and generalize this observation Bounded reward gaps for simple dynamics are important for two reasons:

- **Error Isolation.** Under VP (as well as VE) models, we are able to compute the *exact* reward difference induced by changing $\eta$, free from the well-studied score-approximation and discretization error (Liang et al., 2025), with which we will also analyze in Section 4 and upper-bound reward gaps in those more complex settings.

- **Parametrization Insight.** Parametrization is inherent to the pre-trained (see equation 2) and the fine-tuned model (see equation 9) and affect RLHF performances in practice (Han et al., 2025). Here we show that reward gap is small *even under simple parametrization*. As Section 4 bounds complex controls *quadratic* in $\eta$, our results are valid for general networks.

Our goals are to quantify the reward improvement of fine-tuning $I_\eta$ and the reward gap $\Delta_\eta$ between SDE and ODE inference. Here we formally define:

**Definition 3.2.** *As illustrated in Figure 1, we define*

1. *Improvement of fine-tuning as the reward difference between $\{Y_t^{ODE}\}$ and $\{Y_t^{REF}\}$, $I_\eta :=$ $\mathbb{E}[r(Y_T^{ODE})] - \mathbb{E}[r(Y_T^{REF})] \geq 0$.*

2. *Reward gap as the reward difference between $\{Y_t^{ODE}\}$ and $\{Y_t^{SDE}\}$, $\Delta_\eta :=$ $\left| \mathbb{E}[r(Y_T^{ODE})] - \mathbb{E}[r(Y_T^{SDE})] \right|$.*

In what follows, we analyze VE/VP models with a Gaussian or mixture Gaussian target distribution, where the score function has a closed-form expression without approximation and discretization errors, so the only discrepancy comes from changing $\eta$.

*Remark.* $\mathbb{E}[r(Y_T^{\text{ODE}})] \geq \mathbb{E}[r(Y_T^{\text{REF}})]$ holds for all models, following from the optimality of $\theta_\eta^*$. However, the sign of $\left( \mathbb{E}[r(Y_T^{\text{ODE}})] - \mathbb{E}[r(Y_T^{\text{SDE}})] \right)$ depends on the process dynamics as well as RLHF algorithms.

### 3.1 VE WITH A GAUSSIAN TARGET

We first consider the one-dimensional VE model:

$$dX_t = \sqrt{2t}\, dB_t, \quad \text{with } p_{\text{data}}(\cdot) = \mathcal{N}(0,1). \tag{10}$$

Since $X_t \sim \mathcal{N}(0, t^2 + 1)$, the (exact) score function is $\nabla \log p(t,x) = -\frac{x}{t^2+1}$. So the "pretrained" model is:

$$dY_t^{\text{REF}} = -\frac{(1+\eta^2)(T-t)}{1+(T-t)^2} Y_t^{\text{REF}} dt + \eta\sqrt{2(T-t)} dB_t.$$

Next we set the reward function $r(x) = -(x-1)^2$, so the goal of fine-tuning is to drive the sample towards mean 1. We also specify the fine-tuned SDE and ODE as defined in equation 9 by

$$dY_t^{\text{SDE}} = -\frac{(1+\eta^2)(T-t)}{1+(T-t)^2}(Y_t^{\text{SDE}} + \theta_\eta^*)dt + \eta\sqrt{2(T-t)}dB_t,$$

$$dY_t^{\text{ODE}} = -\frac{T-t}{1+(T-t)^2}(Y_t^{\text{ODE}} + \theta_\eta^*)dt,$$

with $\theta_\eta^*$ maximizing the entropy-regularized reward (equation 8). The following theorem gives bounds for $I_\eta$ and $\Delta_\eta$ under VE, and the proof is deferred to Appendix C.1.

**Theorem 3.1.** *Consider the Variance Exploding model (equation 10), with the reward function*

$$r(x) = -(x-1)^2. \tag{11}$$

*For $\eta > 0$, we have $\theta_\eta^* = -\left[(1+\frac{\beta}{2})\left(1 - (1+T^2)^{-\frac{1+\eta^2}{2}}\right)\right]^{-1}$. Moreover, for $T \geq 1$*

$$0 \leq \Delta_\eta \leq \frac{1}{2T} + o\left(\frac{1}{T}\right) \quad and \quad I_\eta \geq 1 - \frac{1}{2T} + o\left(\frac{1}{T}\right). \tag{12}$$

### 3.2 VP WITH A GAUSSIAN TARGET

Now we consider the one-dimensional VP model:

$$dX_t = -tX_t dt + \sqrt{2t}\,dB_t, \quad \text{with } p_{\text{data}}(\cdot) = \mathcal{N}(0,1). \tag{13}$$

Under the same setup as equation 9 and the VE case, we have:

$$dY_t^{\text{REF}} = -\eta^2(T-t)Y_t^{\text{REF}}dt + \eta\sqrt{2(T-t)}dB_t,$$

$$dY_t^{\text{SDE}} = -\eta^2(T-t)Y_t^{\text{SDE}}\,dt - (1+\eta^2)(T-t)\theta_\eta^*(t)\,dt + \eta\sqrt{2(T-t)}dB_t,$$

$$dY_t^{\text{ODE}} = -(T-t)\theta_\eta^*(t)dt,$$

where a time-dependent control $\theta_\eta(t) := \theta_\eta e^{-\frac{(T-t)^2}{2}}$ is used for fine-tuning. The following theorem gives bounds for $I_\eta$ and $\Delta_\eta$ under VP, and the proof is deferred to Appendix C.2.

**Theorem 3.2.** *Consider the VP model (equation 13), with the reward function $r(x) = -(x-1)^2$. For $\eta > 0$, we have $\theta_\eta^* = -\left[(1+\frac{\beta}{2})\left(1 - e^{-\frac{(1+\eta^2)T^2}{2}}\right)\right]^{-1}$. Moreover, for $T \geq 1$*

$$0 \leq \Delta_\eta \leq \frac{e^{-T^2}}{2} + o\left(e^{-T^2}\right) \quad and \quad I_\eta \geq 1 - \frac{e^{-T^2}}{2} + o\left(e^{-T^2}\right). \tag{14}$$

### 3.3 VE/VP WITH A MIXTURE GAUSSIAN TARGET

The previous results can be extended to multidimensional setting, with a mixture Gaussian target distribution. Recall that the probability density of a mixture Gaussian has the form:

$$\sum_{i=1}^{k} \frac{\alpha_i}{(2\pi)^{d/2}(\det \boldsymbol{\Sigma}_i)^{1/2}} \cdot \exp\left(-\frac{1}{2}(x - \boldsymbol{\mu}_i)^T \boldsymbol{\Sigma}_i(x - \boldsymbol{\mu}_i)\right),$$

where $\alpha_i$ is the weight of the $i$-th Gaussian component. The following corollary bounds the reward gap for a mixture Gaussian target distribution, and the proof is deferred to Appendix C.3.

**Corollary 3.1.** *Let the reward function be $r(\boldsymbol{x}) = -||\boldsymbol{x} - \boldsymbol{r}||^2$ such that $\boldsymbol{\mu}_i \cdot \boldsymbol{r} = 0$ for all $i \in \{1, \ldots, k\}$, $\Sigma_i \equiv \boldsymbol{I}_d$ and $\mathbb{E}[Y_T^{REF}] = \boldsymbol{0}$. Then the same bounds on reward gap hold, i.e.,*

$$\Delta_\eta \leq \begin{cases} (2T)^{-1} & \text{for VE,} \\ e^{-T^2}/2 & \text{for VP.} \end{cases} \tag{15}$$

For all these examples, the reward gap $\Delta_\eta \downarrow 0$ (independent of $\eta$), as $T \to \infty$. Such a phenomenon will also be observed in fine-tuning T2I models with more complex rewards.

| | $\eta$ | ImageReward | PickScore | HPS_v2 | Aesthetic |
|---|---|---|---|---|---|
| **Base** | − | 0.32 | 20.69 | 0.253 | 5.38 |
| **ImageReward** | 1.0 | 0.84 (0.92) | 20.89 (20.94) | **0.268** (**0.271**) | 5.64 (5.72) |
| | 1.2 | **0.92** (**1.03**) | **20.95** (**20.99**) | **0.268** (0.289) | **5.70** (**5.80**) |
| | 1.5 | 0.77 (0.91) | 20.90 (20.95) | 0.266 (0.269) | 5.61 (5.74) |
| **PickScore** | 1.0 | 0.59 (**0.73**) | 21.11 (21.28) | **0.265** (**0.267**) | 5.56 (5.68) |
| | 1.2 | **0.59** (0.69) | **21.17** (21.33) | 0.264 (0.264) | 5.57 (5.69) |
| | 1.5 | 0.57 (0.70) | 21.10 (**21.36**) | 0.264 (0.262) | **5.60** (**5.77**) |

**Table 1:** Performance for ODE (SDE) samplers under DDPO fine-tuning. Bold numbers indicate the highest evaluations among all stochasticity, demonstrating $\eta = 1.2$ in general performs the best.

## 4 NON-VACUOUS BOUND ON THE REWARD GAP

In Section 3, our analysis relies on the explicit computation of score functions with simple controls, which is not available for fine-tuning in general. Here our goal is to bound the Wasserstein-2 distance between $p_{\text{data}}^{\text{SDE}}$ and $p_{\text{data}}^{\text{ODE}}$ with mild assumptions before applying it to a $L_r$-Lipschitz reward $r(\cdot)$, i.e. $|r(y_1) - r(y_2)| \leq L_r \cdot ||y_1 - y_2||$.

**Assumption 4.1.** *The following conditions hold for all $y, y_1, y_2$ along inference trajectories,*

    *1. Lipschitz of $s_\theta$: There exists $L > 0$ such that $||s_\theta(t, y_1) - s_\theta(t, y_2)|| \leq L||y_1 - y_2||$.*

    *2. $L^2$ bound on $s_\theta$: There exists $A > 0$ such that $\sup_{0 \leq t \leq T} \mathbb{E}[||s_\theta(t, y)||^2] \leq A$.*

Condition 1 (Lipschitz of scores) and 2 ($L^2$ bound) are standard in stability analysis. Also, there always exists a continuous-time VP/VE SDE on $[0, 1]$ whose discretization (Song et al., 2021a) matches the large-scale T2I models' (i.e. Stable Diffusion and SDXL) training schedule. In addition, models (FLUX) based on rectified flow inherently uses a normalized time interval. Therefore, for large $\eta$, the following theorem yields $O(\eta^2)$ bound on $W_2$ distance, see proof in Appendix D.1.

**Theorem 4.1.** *If Assumption 4.1 hold with $T = 1$, $W_2(Y_T^{ODE}, Y_T^{SDE}) \leq C \cdot \max\{\eta, \eta^2\} \cdot \sqrt{1 + A}$, where $C$ only depends on $||g||_\infty$ and $L$.*

**Proposition 4.1.** *Let Assumption 4.1 hold with $L_r$-Lipschitz rewards $r$, $\Delta_\eta \leq L_r \cdot C \cdot \max\{\eta, \eta^2\} \cdot \sqrt{1 + A}$, where $C$ only depends on $||g||_\infty$ and $L$.*

An improved $O(\eta)$ bound relies on contractive coefficients (see Appendix D.2). This contractivity can arise from a contractive drift $f$ (Tang & Zhao, 2024). Alternatively, even though the backward dynamics of classical VP models are expansive, a strongly log-concave terminal distribution (Gao et al., 2025) still satisfies the contractive condition (see Assumption D.1), which can hold if the conditional terminal distribution for a fixed prompt $c$ is approximately unimodal.

Quadratic and linear growth rate on $\eta$ is consistent with our empirical observation: T2I fine-tuning preserves quality for ODE inference; rewards only deteriorate under very large $\eta$ (see Table 3).

*Remark.* Discretization errors can be incorporated into the $W_2$ bound as an additional term $\epsilon_d(h)$ depending on the time–step size $h$ (Liang et al., 2025), with $\epsilon_d(h) \to 0$ as $h \to 0$. Under Assumption 4.1, Euler–Maruyama yields such a vanishing term; see Appendix D.3 for details.

## 5 NUMERICAL EXPERIMENTS

In this section, we fine-tune large-scale T2I models under large $\eta$ and examine its reward gap $\Delta_\eta$ with RLHF algorithms DDPO and MixGRPO (see Section 2.3). Preference rewards for fine-tuning DDPO include the LAION aesthetic (Schuhmann et al., 2022), HPS_v2.1 (Wu et al., 2023), PickScore (Kirstain et al., 2023), and ImageReward (Xu et al., 2023). The rewards for fine-tuning MixGRPO include HPS CLIP, PickScore, ImageReward, and Unified Reward (Wang et al., 2025).

According to equation 4 and equation 5, we use a high stochasticity $\eta \geq 1.0$ under **gDDIM scheme** (see Section 2.2 and Appendix E) to generate robust training samples $\{Y_T^{\text{SDE}}\}$ and compare them with deterministic inference samples $\{Y_T^{\text{ODE}}\}$ following equation 9. The stochasticity scale $\eta$ controls the noise level of backward dynamics.

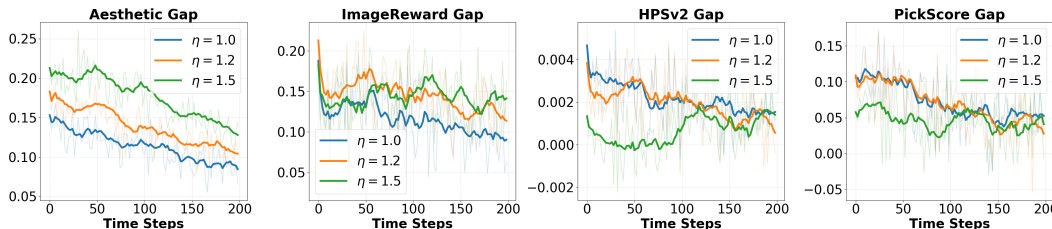

**Figure 3:** Evolution of reward gap (the reward difference between SDE sampling and ODE sampling) during DDPO training under PickScore fine-tuning with stochasticity $\eta \in \{1.0, 1.2, 1.5\}$. The gap for multiple rewards remains small as training step $N$ progresses.

*Remark.* In each experiment, the training stochasticity $\eta$ and the number of training steps $N$ are always specified. Choices of other hyperparameters are detailed in Appendix F.

## 5.1 DDPO

We use Stable Diffusion v1.5 (Rombach et al., 2022) as the base model, and fine-tune it with DDPO. During training, we adopt ImageReward and PickScore as preference rewards, while Aesthetic and HPS_v2 are included as additional evaluation metrics. Figure 7 (and Appendix H) provides representative generations from the fine-tuned models. Our main observations are summarized as follows:

• **High Stochasticity Benefits Moderate Training Steps.** We compare fine-tuning under ImageReward and PickScore at $\eta \in \{1.0, 1.2, 1.5\}$ for $N = 200$ steps. As shown in Table 1, $\eta = 1.2$ under ImageReward achieves the best in-domain and out-of-domain performance, while PickScore's performances depend on evaluation metrics.

• **Decreasing Reward Gap with Quality Improvement.** To study the reward gaps for smaller or larger $N$, we experiment under PickScore with $\eta = 1.2$ and calculate SDE–ODE reward differences (here we report the SDE over ODE performance) under multiple preference functions every 200 steps until reward collapse. As shown in Table 2, the gap decreases as image quality improves for both samplers.

• **Richer Prompt Contents Reduce Reward Gap.** We compare performances with animal versus more comprehensive prompts (see Appendix H.4) under ImageReward with $\eta = 1.2$. As shown in Table 3, complex prompts generate higher post-tuning rewards and higher in-group variance in post-training. Moreover, their richer instructions reduce the SDE–ODE reward gap.

| | $N = 0$ | 200 | 400 | 600 | 800 | 1000 | 1200 | 1400 | (1476) |
|---|---|---|---|---|---|---|---|---|---|
| **ImgRwrd Gap** | 0.160 | 0.119 | 0.102 | 0.028 | 0.057 | 0.027 | **0.006** | 0.020 | (0.564) |
| **HPSv2 Gap** | 0.0047 | 0.0032 | 0.0032 | 0.0018 | 0.0027 | 0.0015 | **-0.0028** | 0.0011 | (0.0308) |
| **Aesthetic Gap** | 0.162 | 0.113 | 0.078 | 0.077 | 0.072 | 0.017 | 0.030 | **-0.010** | (0.685) |
| **PickScore Gap** | 0.115 | 0.146 | 0.167 | 0.118 | 0.178 | 0.106 | 0.129 | **0.090** | (0.907) |
| **SDE Reward** | 20.82 | 21.31 | 21.65 | 21.90 | 22.07 | 22.27 | 22.31 | **22.42** | (18.92) |
| **ODE Reward** | 20.73 | 21.17 | 21.50 | 21.78 | 21.88 | 22.17 | 22.20 | **22.32** | (17.04) |

**Table 2:** PickScore training with $\eta = 1.2$ until reward collapses. Smallest reward gaps and best sampler performances locate at the large training steps $N = 1200, 1400$.

| | **Animal Prompts** | | | **Comprehensive Prompts** | | |
|---|---|---|---|---|---|---|
| | $N = 0$ | 100 | 200 | $N = 0$ | 100 | 200 |
| **Mean** | **0.38** (**0.540**) | 0.67 (0.81) | 0.92 (1.03) | 0.35 (0.47) | **0.76** (**0.87**) | **1.09** (**1.17**) |
| **Std** | 0.81 (0.80) | 0.78 (0.72) | 0.71 (0.65) | **1.03** (**1.00**) | 0.91 (0.87) | 0.80 (0.73) |
| $\Delta_{\eta=1.2}$ | **0.15** | **0.14** | **0.12** | 0.12 | 0.11 | 0.11 |

**Table 3:** Performance Comparison between prompts of different complexity with $\eta = 1.2$. More complicated prompts yields faster fine-tuning improvements, larger in-group variances, and smaller SDE-ODE reward gaps.

## 5.2 MIXGRPO

We use FLUX.1 (Black Forest Labs, 2024) as the base model, and fine-tune it with MixGRPO, which is a sliding-window sampler that alternates between ODE and SDE schemes for 25 training steps in total. Training is carried out with multiple rewards combined using equal weights, while evaluation is reported on ImageReward and HPSClip.

• **Bounded Reward Gap Under High Stochasticity.** As shown in Figure 4, $\Delta_\eta$ converges to zero for both HPS Clip and ImageReward. Also, with $\eta = 1.2$ and ImageReward metric, ODE sampling in MixGRPO consistently outperforms the mixed SDE–ODE scheme, in contrast to the results from DDPO.

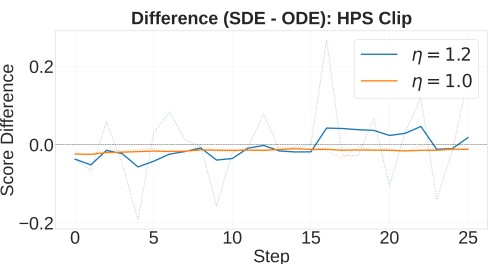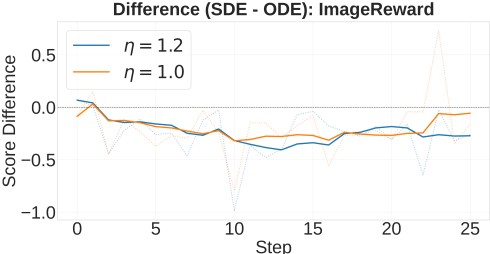

**Figure 4:** Bounded reward gap for MixGRPO, displayed with 7-step moving average. Here a positive score difference means the SDE component achieves higher reward than the ODE component.

• **T2I Quality Improvement.** The upper (evaluated with ImageReward) and lower panels (evaluated with HPS Clip) of Figure 5 further show that samplers trained with larger stochasticity ($\eta = 1.2$) perform consistently better than standard DDIM stochasticity ($\eta = 1.0$) on training prompts. For example, in Figure 6, the generation with $\eta = 1.2$ correctly aligns with the "trapped inside" prompt instruction, whereas the generation with $\eta = 1.0$ fails to do so.

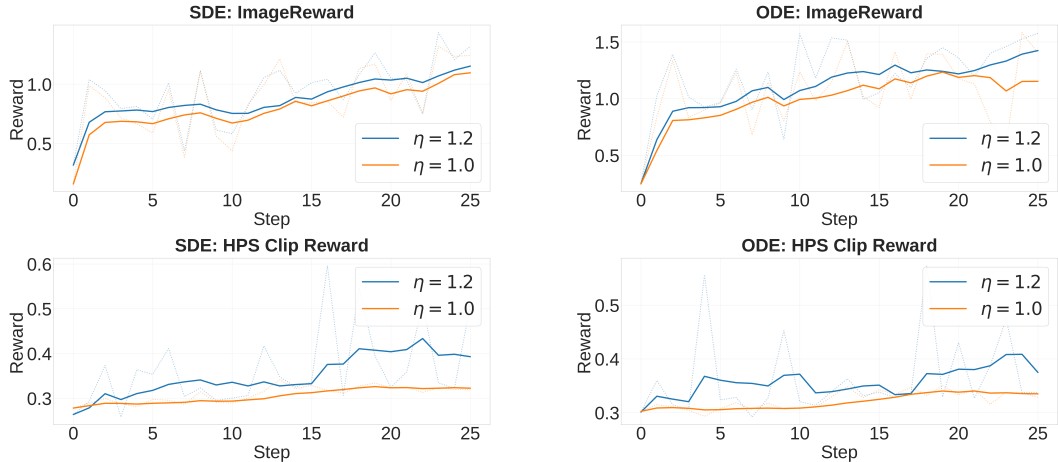

**Figure 5:** Performance improvement for MixGRPO, displayed with 7-step moving average.

## 6 CONCLUSION AND FURTHER DIRECTIONS

This work clarifies the tension between stochastic SDE training and deterministic ODE inference in diffusion RLHF. By proving a bounded reward gap $\Delta_\eta$ and empirically showing that higher training stochasticity (e.g., $\eta = 1.2$) improves deterministic image quality, we provide theoretical support for "training with SDE, inference with ODE". Future work includes quantifying how distribution shift and the choice of reward function separately affect reward gaps.

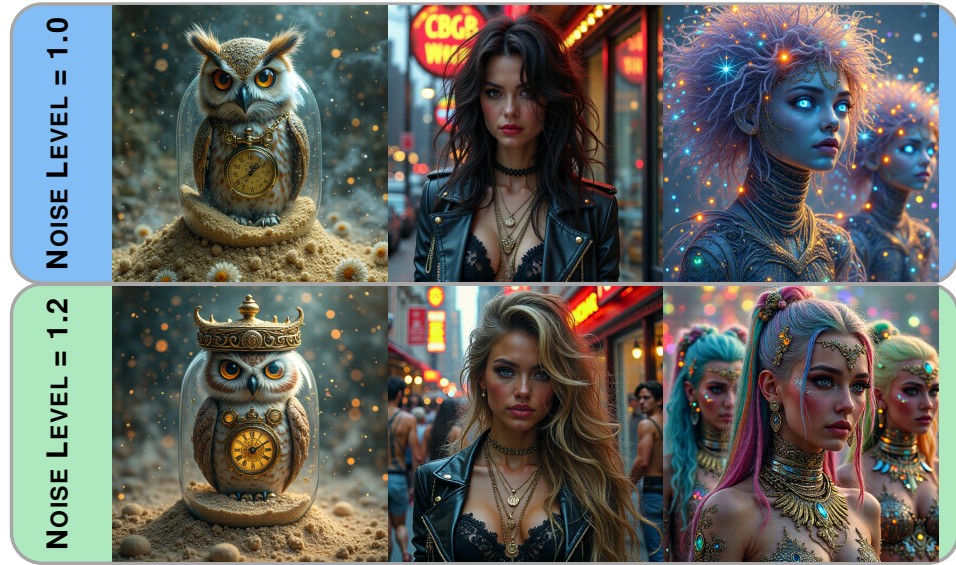

**Figure 6:** Comparison of ODE image generation by FLUX with MixGRPO fine-tuning, stochasticity $\eta = 1.2$ (below) and $\eta = 1.0$ (above). Higher stochasticity shows better alignments to details.
Prompts (from left to right): "A steampunk pocketwatch owl is trapped inside a glass jar buried in sand, surrounded by an hourglass and swirling mist.", "An androgynous glam rocker poses outside CBGB in the style of Phil Hale.", "A digital painting by Loish featuring a rush of half-body, cyberpunk androids and cyborgs adorned with intricate jewelry and colorful holographic dreads."

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

## A    LEMMAS ON LINEAR DYNAMICS WITH GAUSSIAN PRIORS

**Lemma A.1.** *A stochastic process $\{Z_t\}_{t=0}^T$ with first order linear dynamic and initial Gaussian distribution*

$$\begin{cases} dZ_t = f(t)Z_t dt + g(t)dt + h(t)dB_t & t \in [0,T] \\ Z_0 \sim \mathcal{N}(0,1) \end{cases}$$

*is distributed following*

$$Z_t \sim \mathcal{N}\Big(e^{F(t)} \int_0^t e^{-F(s)} g(s)\, ds,\ e^{2F(t)} \int_0^t e^{-2F(t)} h^2(s)\, ds\Big),$$

*in which $F(t)$ is the integrating factor satisfying $F(t) = \int_0^t f(s)ds$.*

**Lemma A.2.** *A stochastic process $\{Z_t\}_{t=0}^T$ with first order linear dynamic and initial Gaussian distribution*

$$\begin{cases} dZ_t = f(t)Z_t dt + g(t)dB_t & t \in [0,T] \\ Z_0 \sim \mathcal{N}(\mu_Z, \sigma_Z^2) \end{cases}$$

*is distributed following*

$$Z_t \sim Z_0 \cdot e^{F(t)} + \mathcal{N}\Big(0, \int_0^t e^{-2F(s)} g^2(s)ds\Big) \cdot e^{F(t)}$$

$$\sim \mathcal{N}\Big(\mu_Z \cdot e^{F(t)}, \Big[\sigma_Z^2 + \int_0^t e^{-2F(s)} g^2(s)ds\Big] \cdot e^{2F(t)}\Big),$$

*in which $F(t)$ is the integrating factor satisfying $F(t) = \int_0^t f(s)ds$.*

**Lemma A.3.** *A parametrized family of stochastic processes $\{Z_t^\theta\}_{t=0}^T$ with initial Gaussian distribution*

$$\begin{cases} dZ_t^\theta = f(t) \cdot (Z_t^\theta + \theta(t))dt + g(t)dB_t & t \in [0,T] \\ Z_0 \sim \mathcal{N}(\mu_Z, \sigma_Z^2) \end{cases}$$

*is distributed following*

$$Z_t \sim Z_0 \cdot e^{F(t)} + \mathcal{N}\Big(\int_0^t e^{-F(s)} f(s)\theta(s)ds, \int_0^t e^{-2F(s)} g^2(s)ds\Big) \cdot e^{F(t)}$$

$$\sim \mathcal{N}\Big(\mu_Z \cdot e^{F(t)} + \int_0^t e^{-F(s)} f(s)\theta(s)ds, \Big[\sigma_Z^2 + \int_0^t e^{-2F(s)} g^2(s)ds\Big] \cdot e^{2F(t)}\Big),$$

*in which $F(t)$ is the integrating factor satisfying $F(t) = \int_0^t f(s)ds$.*

# B   USEFUL PROPOSITIONS

**Proposition B.1.** *The terminal distribution of Variance Exploding parametrized backward dynamics* $\{Y_t^\theta\}_{t=0}^T$ *is always Gaussian following the law*

$$\mathcal{N}(\mu_{Y_T^\theta}, \sigma_{Y_T}^2) := \mathcal{N}\Big(\theta \cdot (1+T^2)^{-\frac{1+\eta^2}{2}} - \theta, 1 - (1+T^2)^{-(1+\eta^2)}\Big).$$

Therefore, reward function (11) takes the form of

$$\mathcal{J}_\eta(\theta) = -\Big(\sigma_{Y_T}^2 + (1 - \mu_{Y_T^\theta})^2\Big).$$

*Proof.* By comparing the coefficients with Lemma A.1, we have

$$\begin{cases} f_\eta(t) = -\frac{(1+\eta^2)(T-t)}{1+(T-t)^2} \\ g_\eta(t) = \eta\sqrt{2(T-t)} \\ \sigma_Z^2 = T^2. \end{cases}$$

Therefore, we first examine the exponential of integrating factor,

$$\begin{aligned} e^{F_\eta(t)} &= \exp\left(\int_0^t f_\eta(s)ds\right) \\ &= \exp\left(\int_0^t -\frac{(1+\eta^2)(T-s)}{1+(T-s)^2}ds\right) \\ &= \exp\left(\int_{1+T^2}^{1+(T-t)^2} \frac{(1+\eta^2)d(1+(T-s)^2)}{2(1+(T-s)^2)}\right) \\ &= \left(\frac{1+T^2}{1+(T-t)^2}\right)^{-\frac{1+\eta^2}{2}}. \end{aligned}$$

At terminal time $T$, the cumulative factor is

$$e^{F_\eta(T)} = \exp\left(\int_0^T f_\eta(s)ds\right) = \left(1+T^2\right)^{-\frac{1+\eta^2}{2}}.$$

Also, the cumulative Gaussian variance generated from the backward process is,

$$\begin{aligned} \int_0^t e^{-2F_\eta(s)}g_\eta^2(s)ds &= \int_0^t \left(\frac{1+T^2}{1+(T-s)^2}\right)^{(1+\eta^2)} \cdot (2\eta^2(T-s))ds \\ &= \int_{1+T^2}^{1+(T-t)^2} (-\eta^2)\left(\frac{1+T^2}{1+(T-s)^2}\right)^{(1+\eta^2)} d(1+(T-s)^2) \\ &= (1+T^2)^{(1+\eta^2)} \int_{1+T^2}^{1+(T-t)^2} (-\eta^2)(1+(T-s)^2)^{-(1+\eta^2)} d(1+(T-s)^2) \\ &= (1+T^2)^{(1+\eta^2)} \cdot \Big((1+(T-t)^2)^{-\eta^2} - (1+T^2)^{-\eta^2}\Big). \end{aligned}$$

At terminal time $T$, the variance from the process is

$$\begin{aligned} \int_0^T e^{-2F_\eta(s)}g_\eta^2(s)ds &= (1+T^2)^{(1+\eta^2)} \cdot \Big(1 - (1+T^2)^{-\eta^2}\Big) \\ &= (1+T^2)^{(1+\eta^2)} - (1+T^2). \end{aligned}$$

Together with the initial Gaussian variance, the terminal distribution remains a zero-mean Gaussian:

$$\begin{aligned} Y_T &\sim \mathcal{N}\Big(0, [(1+T^2)^{(1+\eta^2)} - (1+T^2 - \sigma_Z)] \cdot (1+T^2)^{-(1+\eta^2)}\Big) \\ &\sim \mathcal{N}\Big(0, [(1+T^2)^{(1+\eta^2)} - 1] \cdot (1+T^2)^{-(1+\eta^2)}\Big) \\ &\sim \mathcal{N}\Big(0, 1 - (1+T^2)^{-(1+\eta^2)}\Big). \end{aligned}$$

Now we consider the terminal distribution for the parametrized process $Y_T^\theta$. To reduce the problem to a dynamic with linear drift term, we define

$$Z_t^\theta = Y_t^\theta + \theta.$$

so that

$$\begin{cases} dZ_t = f_\eta(t)Z_t dt + g_\eta(t)dB_t & t \in [0,T] \\ Z_0 \sim \mathcal{N}(\theta, T^2) \end{cases}$$

Observe that both the dynamic variance and the initial distribution variance for $\{Z_t^\theta\}$ and $\{Y_t\}$ are the same, we may directly apply Lemma A.1 to obtain

$$Z_T^\theta \sim \mathcal{N}\big(\theta \cdot (1+T^2)^{-\frac{1+\eta^2}{2}}, 1 - (1+T^2)^{-(1+\eta^2)}\big)$$

Therefore,

$$Y_T^\theta \sim \mathcal{N}\big(\theta \cdot (1+T^2)^{-\frac{1+\eta^2}{2}} - \theta, 1 - (1+T^2)^{-(1+\eta^2)}\big),$$

and thus

$$\begin{cases} \mu_{Y_T^\theta} := \theta \cdot (1+T^2)^{-\frac{1+\eta^2}{2}} - \theta \\ \sigma_{Y_T}^2 := 1 - (1+T^2)^{-(1+\eta^2)} \end{cases} \tag{16}$$

In addition, we can now give a closed form representation of our reward,

$$\begin{aligned} \mathcal{J}_\eta(\theta) &= \mathbb{E}[(Y_T^\theta - 1)^2] \\ &= \mathbb{E}[Y_T^2] - 2\mathbb{E}[Y_T] + 1 \\ &= \sigma_{Y_T}^2 + \mu_{Y_T^\theta}^2 - 2\mu_{Y_T^\theta} + 1 \\ &= \sigma_{Y_T}^2 + \big(1 - \mu_{Y_T^\theta}\big)^2, \end{aligned}$$

which yields to the desired expression. $\qquad\square$

**Proposition B.2.** *Given $\beta, \eta, T$, the unique maximizer to the entropy regularized target (8) is:*

$$\theta_\eta^* = -\Big(\big(1 + \frac{\beta}{2}\big) \cdot \big[1 - (1+T^2)^{-\frac{1+\eta^2}{2}}\big]\Big)^{-1}.$$

*Proof.* A classical distance result on two Gaussian distributions (Hershey & Olsen, 2007) states:

**Lemma B.1.** *The KL divergence for two Gaussian distributions $P \sim \mathcal{N}(\mu_1, \sigma_1)$ and $Q \sim \mathcal{N}(\mu_2, \sigma_2)$,*

$$\mathrm{KL}(P\|Q) = \log\Big(\frac{\sigma_2}{\sigma_1}\Big) + \frac{\sigma_1^2 + (\mu_1 - \mu_2)^2}{2\sigma_2^2} - \frac{1}{2}.$$

In our model, $P \sim Y_T^\theta$ and $Q \sim Y_T^\odot$, so $\mu_1 = \mu_{Y_T^\theta}, \sigma_1 = \sigma_{Y_T}, \mu_2 = 0, \sigma_2 = 1$.

$$\begin{aligned} \mathrm{KL}(Y_T^\theta \| Y_T^\odot) &= \log(\frac{1}{\sigma_{Y_T}}) + \frac{\sigma_{Y_T}^2 + \mu_{Y_T^\theta}^2 - 1}{2} \\ &= -\log(\sigma_{Y_T}) + \frac{\sigma_{Y_T}^2 + \mu_{Y_T^\theta}^2 - 1}{2}. \end{aligned}$$

Therefore,

$$\begin{aligned} -F_\eta(\theta) &= \sigma_{Y_T}^2 + \big(\mu_{Y_T^\theta} - 1\big)^2 + \big(-\beta\log(\sigma_{Y_T}) + \frac{\beta}{2} \cdot (\sigma_{Y_T}^2 + \mu_{Y_T^\theta}^2 - 1)\big) \\ &= \big(\sigma_{Y_T}^2 - \beta\log(\sigma_{Y_T}) + \frac{\beta}{2}\sigma_{Y_T}^2 - \frac{\beta}{2}\big) + \big(\mu_{Y_T^\theta} - 1\big)^2 + \frac{\beta}{2}\mu_{Y_T^\theta}^2 \\ &= \big(\sigma_{Y_T}^2 - \beta\log(\sigma_{Y_T}) + \frac{\beta}{2}\sigma_{Y_T}^2 - \frac{\beta}{2} + 1\big) + \big(1 + \frac{\beta}{2}\big)\mu_{Y_T^\theta}^2 - 2\mu_{Y_T^\theta}. \end{aligned}$$

So it suffices to minimize a quadratic function w.r.t $\mu_{Y_T^\theta}$, of which we know

$$\mu_{Y_T^{\theta*}} = -\frac{-2}{2(1+\frac{\beta}{2})} = (1+\frac{\beta}{2})^{-1};$$

and thus

$$\theta_\eta^* \cdot \left[(1+T^2)^{-\frac{1+\eta^2}{2}} - 1\right] = \mu_{Y_T^{\theta*}} = (1+\frac{\beta}{2})^{-1}.$$

This gives us the unique maximizer

$$\theta_\eta^* = \left((1+\frac{\beta}{2}) \cdot \left[(1+T^2)^{-\frac{1+\eta^2}{2}} - 1\right]\right)^{-1}$$

as desired. $\qquad\square$

**Proposition B.3.** *The terminal distribution of Variance Preserving parametrized backward dynamics $\{Y_t^\theta\}_{t=0}^T$ is always Gaussian following the law*

$$\mathcal{N}(\mu_{Y_T^\theta}, 1) := \mathcal{N}\left(\theta \cdot e^{-\frac{(1+\eta^2)\cdot T^2}{2}} - \theta, 1\right).$$

*And reward function (11) takes the form of $\mathcal{J}_\eta(\theta) = -\left(1 + (1 - \mu_{Y_T^\theta})^2\right)$.*

*Proof.* By comparing with the coefficients in Lemma A.2, we have

$$\begin{cases} f_\eta(t) = -\eta^2(T-t) \\ g_\eta(t) = -(1+\eta^2)(T-t)e^{-\frac{(T-t)^2}{2}}\theta_\eta^* \\ h_\eta(t) = \eta\sqrt{2(T-t)} \end{cases}$$

Therefore, we first examine the exponential of integrating factor,

$$e^{F_\eta(t)} = \exp\left(\int_0^t f_\eta(s)ds\right)$$

$$= \exp\left(-\eta^2 \int_{T-t}^T s\,ds\right)$$

$$= \exp\left(-\frac{\eta^2}{2} \cdot \left(T^2 - (T-t)^2\right)\right).$$

At terminal time $T$, the cumulative factor is

$$e^{F_\eta(T)} = \exp\left(-\frac{\eta^2}{2} \cdot \left(T^2 - (T-T)^2\right)\right) = e^{-\frac{\eta^2 T^2}{2}}.$$

Now we are able to compute

$$\int_0^t e^{-F_\eta(s)}g(s)\,ds = \int_0^t e^{\frac{\eta^2}{2}\left(T^2-(T-s)^2\right)} \cdot \left(-(1+\eta^2)(T-s)e^{-\frac{(T-s)^2}{2}}\theta_\eta^*\right)ds$$

$$= (1+\eta^2) \cdot \theta_\eta^* \cdot \int_{T-t}^T e^{\frac{\eta^2}{2}\left(T^2-s^2\right)} \cdot \left(se^{-\frac{s^2}{2}}\right)ds$$

$$= (1+\eta^2) \cdot \theta_\eta^* \cdot \int_{T-t}^T e^{\frac{\eta^2 T^2}{2}} \cdot \left(se^{-\frac{(1+\eta^2)s^2}{2}}\right)ds$$

$$= (1+\eta^2) \cdot \theta_\eta^* \cdot e^{\frac{\eta^2 T^2}{2}} \cdot \left[\frac{1}{1+\eta^2} \cdot e^{-\frac{(1+\eta^2)s^2}{2}}\right]_{s=T-t}^{s=T}$$

$$= \theta_\eta^* \cdot e^{\frac{\eta^2 T^2}{2}} \cdot \left(e^{-\frac{(1+\eta^2)T^2}{2}} - e^{-\frac{(1+\eta^2)(T-t)^2}{2}}\right)$$

Therefore,

$$\mu_{Y_T^\theta} = e^{-\frac{\eta^2 T^2}{2}} \cdot \theta_\eta^* \cdot e^{\frac{\eta^2 T^2}{2}} \cdot (e^{-\frac{(1+\eta^2)T^2}{2}} - e^0) = \theta_\eta^* \cdot (e^{-\frac{(1+\eta^2)T^2}{2}} - 1)$$

To the variance preserving property, it suffices to show

$$\frac{1}{e^{2F_\eta(t)}} = \int_0^t e^{-2F_\eta(t)} h_\eta^2(s)\, ds.$$

In fact,

$$\frac{d}{dt} e^{-2F_\eta(t)} = e^{-2F_\eta(t)} \cdot \frac{d(-\eta^2(T-t)^2)}{dt} = e^{-2F_\eta(t)} \cdot \left(\eta^2 \cdot 2(T-t)\right) = e^{-2F_\eta(t)} \cdot h_\eta^2(t).$$

$\square$

**Proposition B.4.** *Given $\beta, \eta, T$, the unique maximizer to the entropy regularized target (8) is:*

$$\theta_\eta^* = -\left((1+\frac{\beta}{2}) \cdot \left[1 - e^{-\frac{(1+\eta^2)\cdot T^2}{2}}\right]\right)^{-1}.$$

*Proof.* Since $\sigma \equiv 1$, according to Lemma B.1, the maximum reward is attained at

$$\mu_{Y_T^{\theta*}} = -\frac{-2}{2(1+\frac{\beta}{2})} = (1+\frac{\beta}{2})^{-1}.$$

By Proposition B.1,

$$\theta_\eta^* \cdot \left[e^{-\frac{(1+\eta^2)\cdot T^2}{2}} - 1\right] = \mu_{Y_T^{\theta*}} = (1+\frac{\beta}{2})^{-1}.$$

This gives us the unique maximizer

$$\theta_\eta^* = \left((1+\frac{\beta}{2}) \cdot \left[e^{-\frac{(1+\eta^2)\cdot T^2}{2}} - 1\right]\right)^{-1}.$$

$\square$

## C  PROOFS FOR SECTION 3

### C.1  PROOF OF THEOREM 3.1

We first consider the $Y_T^{ODE}$ process as discussed in Section 3.1. Since $\eta = 0$, by Proposition B.2,

$$\mu_{Y_T^{ODE}} = \theta_\eta^* \cdot (1 + T^2)^{-\frac{1}{2}} - \theta_\eta^* = (1 + \frac{\beta}{2})^{-1} \cdot \frac{(1 + T^2)^{-\frac{1}{2}} - 1}{(1 + T^2)^{-\frac{1+\eta^2}{2}} - 1},$$

and

$$\sigma_{Y_T^{ODE}}^2 = \sigma_{Y_T^0}^2 = 1 - (1 + T^2)^{-1}.$$

Moreover, the quadratic reward $\mathcal{J}_{ODE}$ for $Y_T^{ODE}$ is

$$\mathcal{J}_0(\theta_\eta^*) = -\left(\sigma_{Y_T^{ODE}}^2 + (\mu_{Y_T^{ODE}} - 1)^2\right)$$

$$= (-1) + (1 + T^2)^{-1} - \left(1 - (1 + \frac{\beta}{2})^{-1} \cdot \frac{(1 + T^2)^{-\frac{1}{2}} - 1}{(1 + T^2)^{-\frac{1+\eta^2}{2}} - 1}\right)^2.$$

Similarly, reward $\mathcal{J}_{SDE}$ for $Y_T^{SDE}$ is

$$\mathcal{J}_\eta(\theta_\eta^*) = -\left(\sigma_{Y_T^{SDE}}^2 + (\mu_{Y_T^{SDE}} - 1)^2\right)$$

$$= (-1) + (1 + T^2)^{-(1+\eta^2)} - \left(1 - (1 + \frac{\beta}{2})^{-1}\right)^2.$$

For simplification, we denote

$$\bar{T} := 1 + T^2 \in (T^2, 2T^2), \quad \bar{\beta} := (1 + \frac{\beta}{2})^{-1} \in (0, 1].$$

Now the reward gap

$$\Delta_\eta = \mathcal{J}_{SDE} - \mathcal{J}_{ODE}$$

$$= \left((1 + T^2)^{-(1+\eta^2)} - (1 + T^2)^{-1}\right)$$

$$- \left(\left(1 - (1 + \frac{\beta}{2})^{-1}\right)^2 - \left(1 - (1 + \frac{\beta}{2})^{-1} \cdot \frac{(1 + T^2)^{-\frac{1}{2}} - 1}{(1 + T^2)^{-\frac{1+\eta^2}{2}} - 1}\right)^2\right)$$

$$= \left(\frac{\bar{T}^{-\eta^2} - 1}{\bar{T}}\right) - \left((1 - \bar{\beta})^2 - \left(1 - \bar{\beta} \cdot \frac{\bar{T}^{-\frac{1}{2}} - 1}{\bar{T}^{-\frac{1+\eta^2}{2}} - 1}\right)^2\right).$$

Since $0 < \bar{T}^{-\eta^2} \leq 1$, we can bound

$$-\bar{T}^{-1} < \frac{\bar{T}^{-\eta^2} - 1}{\bar{T}} \leq 0,$$

and

$$1 - \bar{T}^{-\frac{1}{2}} = \frac{\bar{T}^{-\frac{1}{2}} - 1}{0 - 1} < \frac{\bar{T}^{-\frac{1}{2}} - 1}{\bar{T}^{-\frac{1+\eta^2}{2}} - 1} \leq \frac{\bar{T}^{-\frac{1}{2}} - 1}{\bar{T}^{-\frac{1}{2}} - 1} = 1.$$

Therefore,

$$|\Delta_\eta| = \left|\frac{\bar{T}^{-\eta^2} - 1}{\bar{T}}\right| + \left|(1 - \bar{\beta})^2 - \left(1 - \bar{\beta} \cdot \frac{\bar{T}^{-\frac{1}{2}} - 1}{\bar{T}^{-\frac{1+\eta^2}{2}} - 1}\right)^2\right|$$

$$\leq \bar{T}^{-1} + \left|(1 - \bar{\beta})^2 - \left((1 - \bar{\beta}) + \bar{\beta} \cdot \bar{T}^{-\frac{1}{2}}\right)^2\right|$$

$$\leq \bar{T}^{-1} + \left(\bar{\beta}^2 \cdot \bar{T}^{-1} + 2\bar{\beta}(1 - \bar{\beta})\bar{T}^{-\frac{1}{2}}\right)$$

$$\leq 2\bar{\beta}(1 - \bar{\beta})T^{-1} + o(T^{-1})$$

$$\leq \frac{1}{2T} + o\left(\frac{1}{T}\right).$$

Now we attempt to bound the reference gap. The reward $\mathcal{J}_{REF}$ for $Y_T^{REF}$ is

$$\mathcal{J}_\eta(0) = -\left(\sigma^2_{Y_T^{REF}} + (\mu_{Y_T^{REF}} - 1)^2\right)$$

$$= (-1) + (1 + T^2)^{-(1+\eta^2)} - 1.$$

Finally we have,

$$\Delta_\eta^{REF} = \mathcal{J}_{ODE} - \mathcal{J}_{REF}$$

$$= \bar{T}^{-1} - \left(1 - \bar{\beta} \cdot \frac{\bar{T}^{-\frac{1}{2}} - 1}{\bar{T}^{-\frac{1+\eta^2}{2}} - 1}\right)^2 - \left(\bar{T}^{-(1+\eta^2)} - 1\right)$$

$$\geq \bar{T}^{-1} - \left(2\bar{\beta}(1 - \bar{\beta})\bar{T}^{-\frac{1}{2}} + o(\bar{T}^{-\frac{1}{2}})\right) - \left(\bar{T}^{-1} - 1\right)$$

$$\geq 1 - \frac{1}{2T} + o\left(\frac{1}{T}\right)$$

## C.2   PROOF OF THEOREM 3.2

Similar to Appendix C.1, the quadratic reward (11) for $Y_T^{ODE}$ is

$$\mathcal{J}_{ODE} = -\left(\sigma^2_{Y_T^{ODE}} + (\mu_{Y_T^{ODE}} - 1)^2\right)$$

$$= (-1) - \left(1 - (1 + \frac{\beta}{2})^{-1} \cdot \frac{e^{-\frac{T^2}{2}} - 1}{e^{-\frac{(1+\eta^2)T^2}{2}} - 1}\right)^2.$$

And the reward for $Y_T^{SDE}$ is

$$\mathcal{J}_{SDE} = -\left(\sigma^2_{Y_T^{SDE}} + (\mu_{Y_T^{SDE}} - 1)^2\right)$$

$$= (-1) - \left(1 - (1 + \frac{\beta}{2})^{-1}\right)^2.$$

With $\bar{\beta} := (1 + \frac{\beta}{2})^{-1} \in (0, 1]$,

$$|\Delta_\eta| = \left|(1 - \bar{\beta})^2 - \left(1 - \bar{\beta} \cdot \frac{e^{-\frac{T^2}{2}} - 1}{e^{-\frac{(1+\eta^2)T^2}{2}} - 1}\right)^2\right|$$

$$\leq \left|(1 - \bar{\beta})^2 - \left((1 - \bar{\beta}) + \bar{\beta} \cdot e^{-\frac{T^2}{2}}\right)^2\right|$$

$$\leq \frac{e^{-\frac{T^2}{2}}}{2} + o\left(e^{-T^2}\right).$$

Finally, $Y_T^{REF} \sim \mathcal{N}(0, 1)$, so

$$\mathcal{J}_{REF} = (-1) - (1 - 0)^2.$$

And thus

$$\Delta_\eta^{REF} = \mathcal{J}_{ODE} - \mathcal{J}_{REF}$$

$$= 1 - \left(1 - (1 + \frac{\beta}{2})^{-1} \cdot \frac{e^{-\frac{T^2}{2}} - 1}{e^{-\frac{(1+\eta^2)T^2}{2}} - 1}\right)^2$$

$$\geq 1 - \frac{e^{-\frac{T^2}{2}}}{2} + o(e^{-T^2})$$

### C.3 Proof of Corollary 3.1

Let $\bullet \in \{SDE, ODE\}$. We decompose $Y^{\theta,\bullet} := Y^{\theta,\bullet}_{\parallel} + Y^{\theta,\bullet}_{\perp}$ in terms of $\boldsymbol{r}$. Therefore,

$$r(Y^{\theta,\bullet}(T)) = -\left|Y^{\theta,\bullet}_{\perp} + (Y^{\theta,\bullet}_{\parallel} - \boldsymbol{r})\right|^2 = -\left(\left|Y^{\theta,\bullet}_{\perp}\right|^2 + \left|(Y^{\theta,\bullet}_{\parallel} - \boldsymbol{r})\right|^2\right)$$

Since $\boldsymbol{\mu}_i \perp \boldsymbol{r}$, $\mathbb{E}[Y^{\theta}_{\parallel}(0)] = 0$. Also, the backward dynamic injects a scaled multiple of $\boldsymbol{I}_d$ noise, so the coordinate-wise dynamics are independent. Therefore, we are able to analyze the $\mathrm{Span}(\boldsymbol{r})$ subspace via separating each dimension $\boldsymbol{e}_k \in \mathrm{Span}(\boldsymbol{r})$, in which $\left\{\mathrm{Proj}_{\boldsymbol{e}_k}(Y^{\theta^*,SDE}_{\parallel})\right\}$ and $\left\{\mathrm{Proj}_{\boldsymbol{e}_k}(Y^{\theta^*,ODE}_{\parallel})\right\}$ follows a similar controlled motion as discussed in Theorem 3.1 and 3.2. Moreover, the score function can be bounded by a constant of $\max_i \sigma_i, \min_i^{-1} \sigma_i$ (Liang et al., 2025).

On the other hand, $\theta_{\perp} = \boldsymbol{0}$ is a minimizer for $\left|Y^{\theta,\bullet}_{\perp}\right|^2$, since an additional drift perpendicular to $\boldsymbol{r}$ does not alter reward variance but pushes away reward mean of $Y^{\theta,\bullet}_{\perp}(T)$ from reference priors. Therefore, $\theta = \theta_{\parallel}$. A similar analysis on Gaussian priors with controlled drifts for VP and VE dynamics yields to the desired bounds.

## D Proof for Section 4

**Proposition D.1.** *(Gronwall's) Suppose integrable functions* $u, \alpha, \beta : [0, T] \to \mathbb{R}$ *satisfies* $u'(t) \leq \alpha(t)u(t) + \beta(t)$ ,

$$u(T) \leq e^{\int_0^T \alpha(s)ds} \left(u(0) + \int_0^T e^{-\int_0^s \alpha(\tau)d\tau}\beta(s)ds\right). \tag{17}$$

A proof can be found in (Evans, 2010).

### D.1 Proof of Theorem 4.1

Let $u(t) := \mathbb{E}\left[||Y^{ODE}_t - Y^{SDE}_t||^2\right]$. Note that $u(0) = 0$. Therefore, with appropriate $\alpha, \beta$ satisfying the conditions of equation 17,

$$u(T) \leq e^{\alpha T}\left(\int_0^T \beta \cdot e^{-\alpha s}\,ds\right) = \frac{(e^{\alpha T} - 1)\cdot\beta}{\alpha}. \tag{18}$$

By Ito's Lemma,

$$\begin{aligned}
u'(t) &= \frac{d}{dt}\mathbb{E}\left[||Y^{\mathrm{ODE}}_t - Y^{\mathrm{SDE}}_t||^2\right] \\
&= 2\mathbb{E}\big\langle Y^{\mathrm{ODE}}_t - Y^{\mathrm{SDE}}_t, -f(t, Y^{\mathrm{ODE}}_t) + f(t, Y^{SDE}_t)\big\rangle \\
&\quad + g^2(t)\mathbb{E}\big\langle Y^{\mathrm{ODE}}_t - Y^{\mathrm{SDE}}_t, s_\theta(t, Y^{\mathrm{ODE}}_t) - s_\theta(t, Y^{\mathrm{SDE}}_t)\big\rangle \\
&\quad - g^2(t)\mathbb{E}\big\langle Y^{\mathrm{ODE}}_t - Y^{\mathrm{SDE}}_t, \eta^2 \cdot s_\theta(t, Y^{\mathrm{SDE}}_t)\big\rangle + \eta^2 g^2(t).
\end{aligned}$$

According to (Tang & Zhao, 2025),

$$\mathbb{E}\big\langle Y^{\mathrm{ODE}}_t - Y^{\mathrm{SDE}}_t, -f(t, Y^{\mathrm{ODE}}_t) + f(t, Y^{\mathrm{SDE}}_t)\big\rangle = \begin{cases} 0 & \text{for VE,} \\ g^2(t)u(t)/2 & \text{for VP.} \end{cases}$$

By Condition 1,

$$\begin{aligned}
\mathbb{E}\big\langle Y^{\mathrm{ODE}}_t - Y^{\mathrm{SDE}}_t, s_\theta(t, Y^{\mathrm{ODE}}_t) - s_\theta(t, Y^{\mathrm{SDE}}_t))\big\rangle &\leq \mathbb{E}\big\langle Y^{\mathrm{ODE}}_t - Y^{\mathrm{SDE}}_t, L \cdot \big(Y^{\mathrm{ODE}}_t - Y^{\mathrm{SDE}}_t\big)\big\rangle \\
&= L \cdot u(t).
\end{aligned}$$

By Condition 2,

$$\left|\mathbb{E}\left\langle Y_t^{\text{ODE}} - Y_t^{\text{SDE}}, \eta^2 \cdot s_\theta(t, Y_t^{\text{SDE}}) \right\rangle\right| \leq u(t) + \frac{\eta^4}{4} \cdot \mathbb{E}[||s_\theta(t, Y_t^{\text{SDE}})||^2]$$

$$= u(t) + \frac{\eta^4}{4} \cdot A.$$

Together, by absorbing $||g||_\infty$ into an $\eta$-free constant $C_{\text{scheme}}$,

$$u'(t) \leq ||g||_\infty^2 u(t) + ||g||_\infty^2 \cdot L \cdot u(t) + ||g||_\infty^2 \left(u(t) + \frac{\eta^4}{4} A\right) + ||g||_\infty^2 \cdot \eta^2$$

$$\leq ||g||_\infty^2 u(t) + ||g||_\infty^2 \cdot L \cdot u(t) + ||g||_\infty^2 \left(u(t) + \frac{\eta^4}{4} A\right) + ||g||_\infty^2 \cdot \eta^2$$

$$\leq \underbrace{(C_{\text{scheme}} \cdot L)}_{\alpha} \cdot u(t) + \underbrace{\max\{\eta^4, \eta^2\} \cdot \left(\frac{A}{4} + 1\right) \cdot ||g||_\infty^2}_{\beta}.$$

And we apply equation 18 with $T = 1$:

$$u(T) \leq \frac{(e^{\alpha T} - 1) \cdot \beta}{\alpha}$$

$$\leq \frac{\exp(C_{\text{scheme}} \cdot L)}{C_{\text{scheme}} \cdot L} \cdot \max\{\eta^4, \eta^2\} \cdot \left(\frac{A}{4} + 1\right).$$

Finally, the $W_2$ distance can be bounded in terms of this $L^2$–distance via the chosen coupling:

$$W_2\left(\mathcal{L}(Y_T^{\text{ODE}}), \mathcal{L}(Y_T^{\text{SDE}})\right) \leq \left(\mathbb{E}\left[||Y_T^{\text{ODE}} - Y_T^{\text{SDE}}||^2\right]\right)^{1/2} \leq C(||g||_\infty, L) \max\{\eta, \eta^2\} \sqrt{1 + A}.$$

## D.2 PROOF FOR STRONGLY LOG-CONCAVE DISTRIBUTIONS

**Assumption D.1.** *(Strong log-concavity): Exists $\kappa > 1$ such that for all $t, y_1, y_2$:*

$$\langle y_1 - y_2, s_\theta(t, y_1) - s_\theta(t, y_2) \rangle \leq -\kappa ||y_1 - y_2||^2.$$

**Theorem D.1.** *If Assumption 4.1 and D.1 hold, $W_2(Y_T^{SDE}, Y_T^{ODE}) \leq \eta ||g||_\infty \frac{\sqrt{A + 2\kappa - 2}}{\kappa - 1}$.*

*Proof.* With the additional log-concavity assumption on the score function,

$$\mathbb{E}\left\langle Y_t^{\text{ODE}} - Y_t^{\text{SDE}}, s_\theta(t, Y_t^{\text{ODE}}) - s_\theta(t, Y_t^{\text{SDE}})) \right\rangle \leq -\kappa \cdot u(t).$$

In this case, the coefficient $\alpha$ becomes *contractive* as we may take $\delta = \frac{\kappa - 1}{2\kappa}$:

$$u'(t) \leq ||g||_\infty^2 u(t) - \kappa \cdot ||g||_\infty^2 \cdot u(t) + \eta^2 ||g||_\infty^2 \left(\delta \cdot \kappa \cdot u(t) + \frac{A}{4\delta \cdot \kappa}\right) + ||g||_\infty^2 \cdot \eta^2$$

$$\leq \underbrace{-\kappa ||g||_\infty^2 \left(1 - \frac{1}{\kappa} - \delta\right)}_{\alpha} \cdot u(t) + \underbrace{\eta^2 ||g||_\infty^2 \left(1 + \frac{A}{4\delta \cdot \kappa}\right)}_{\beta}.$$

For arbitrary time horizon $T$, equation 18 gives:

$$u(T) \leq \frac{(1 - e^{-\alpha T}) \cdot \beta}{-\alpha}$$

$$\leq \frac{1}{\kappa \left(1 - \frac{1}{\kappa} - \delta\right)} \cdot \eta^2 ||g||_\infty^2 \left(1 + \frac{A}{4\delta \cdot \kappa}\right)$$

$$= \eta^2 ||g||_\infty^2 \cdot \frac{2(\kappa - 1) + A}{(\kappa - 1)^2}.$$

$\square$

*Remark.* Assumption 4.2 is valid for a *unimodal* terminal distribution, in which a strongly log-concave coefficient $\kappa(T)$ satisfies

$$\mathbb{E}\big\langle Y_T^{\mathrm{ODE}} - Y_T^{\mathrm{SDE}}, s_\theta(T, Y_T^{\mathrm{ODE}}) - s_\theta(T, Y_T^{\mathrm{SDE}}) \big\rangle \leq -\kappa(T) \cdot u(T).$$

In intermediate time steps,

$$\mathbb{E}\big\langle Y_t^{\mathrm{ODE}} - Y_t^{\mathrm{SDE}}, s_\theta(t, Y_t^{\mathrm{ODE}}) - s_\theta(t, Y_t^{\mathrm{SDE}})) \big\rangle \leq -\kappa(t) \cdot u(t),$$

in which (Tang & Zhao, 2025) bounds

$$\kappa(t) \geq \frac{\kappa(T)}{\exp\left(-\int_0^{T-t} f(s)\, ds\right) + \kappa(T) \cdot \int_0^{T-t} \exp\left(-\int_s^{T-t} f(\tau) d\tau\right) ds}.$$

Since $\kappa(t)$ is strictly positive on $[0, T]$, a global coefficient $\kappa_{\mathrm{global}} = \inf_t \kappa(t)$ exists.

### D.3 DISCRETIZATION ERROR ANALYSIS

Let $N$ be the number of time steps and $h = T/N$ be the step size, we consider the DDPM sampling scheme with Euler–Maruyama discretization. Our Assumption 4.1 are equivalent to Assumptions 6.1 and 6.2 ($L$-Lipschitz score and finite second moment) in (Liang et al., 2025), which develops from (Chen et al., 2023a). For $h \lesssim 1/L$,

$$\mathrm{TV}(q_T^h, p_0) \lesssim \underbrace{\sqrt{\mathrm{KL}(p_0\|\mathcal{N}(0,I))}e^{-T}}_{\text{forward convergence}} + \underbrace{(L\sqrt{dh} + Lm_2 h)\sqrt{T}}_{\text{discretization error}} + \underbrace{\varepsilon_0\sqrt{T}}_{\text{score error}},$$

where $q_T^h$ is the law of the discrete sampler at time $T$, $d$ is the ambient space dimension, and $m_2^2 = \mathbb{E}_{p_0}\|X\|^2$. In particular, the discretization contribution to the total variation distance is $O(h^{1/2})$ as $h \to 0$ for fixed $T, d, L, m_2$.

Denote $Y_T^{\mathrm{ODE},h}$ and $Y_T^{\mathrm{SDE},h}$ as the ODE/SDE fine-tuned inference under step size $h$, by triangular inequality,

$$W_2(Y_T^{\mathrm{ODE},h}, Y_T^{\mathrm{SDE},h}) \leq W_2(Y_T^{\mathrm{ODE}}, Y_T^{\mathrm{SDE}}) + \underbrace{W_2(Y_T^{\mathrm{ODE},h}, Y_T^{\mathrm{ODE}}) + W_2(Y_T^{\mathrm{SDE}}, Y_T^{\mathrm{SDE},h})}_{\epsilon_d(h)}$$

Since $\sup_t \mathbb{E}\|X_t\|^2 \leq \infty$, a standard coupling method between $W_2$ and TV distance gives.

$$W_2(Y_T^\theta, Y_T^{\theta,h}) \lesssim \mathrm{TV}(Y_T^\theta, Y_T^{\theta,h})^{1/2}.$$

Therefore, the contribution of time discretization to the $W_2$ error is bounded by the additive term

$$\epsilon_d(h) := C\left(L\sqrt{dh} + Lm_2 h\right)^{1/2},$$

for some constant $C$ depending only on $T, d$. In the continuous time limit, $\epsilon_d(h) \to 0$.

## E DISCUSSION ON DDIM AND GDDIM UNDER HIGH STOCHASTICITY

When $1.0 < \eta < +\infty$, the stochasticity level of gDDIM is always upper-bounded by

$$\sigma_t^{\mathrm{gDDIM}}(\eta) = (1 - \alpha_{t-\Delta t})\left(1 - \left(\frac{1 - \alpha_{t-\Delta t}}{1 - \alpha_t}\right)^{\eta^2}\left(\frac{\alpha_t}{\alpha_{t-\Delta t}}\right)^{\eta^2}\right) \leq 1 - \alpha_{t-\Delta t} < +\infty;$$

whereas the linear interpolation of DDIM gives an unbounded

$$\sigma_t^{\mathrm{DDIM}}(\eta) = \eta\sqrt{\frac{1 - \alpha_{t-\Delta t}}{1 - \alpha_t}}\sqrt{1 - \frac{\alpha_t}{\alpha_{t-\Delta t}}}.$$

In order to ensure the well-posedness of $\sqrt{(1 - \alpha_{t-\Delta t} - \sigma_t^2)(1 - \alpha_t)}$ in equation 4, the conventional discretization requires

$$\sigma_t^{\mathrm{DDIM}} \leq 1 - \alpha_{t-\Delta t}.$$

We may attempt to bypass the restriction by regulating

$$\sigma_t^{\mathrm{linear\ DDIM}}(\eta) = \min\{1 - \alpha_{t-\Delta t}, \eta_t^{\mathrm{DDIM}}(\eta)\}.$$

However, this changes the terminal marginals in equation 3. Therefore, the classical DDIM interpolation cannot simultaneously support arbitrary $\eta > 1.0$ and preserve the terminal marginal, whereas gDDIM does.

## F  HYPERPARAMETERS

All experiments are conducted on 7 Nvidia A100 GPUs. Mixed precision training is used with the bfloat16 (bf16) format.

### F.1  DDPO EXPERIMENTS

We follow the setup of Black et al. (2024), using denoising step $T = 50$ and guidance weight $w = 5.0$ throughout all experiments. We also use the AdamW optimizer Loshchilov & Hutter (2019) with default weight decay 1e-4 and optimal learning rates for different reward functions. Reward gaps under four reward functions are shown in Figure 3 in Section 5 and Appendix H.

**Table 4:** DDPO hyperparameters

|      |                                | ImageReward | PickScore | HPSv2 | Aesthetic |
|------|--------------------------------|-------------|-----------|-------|-----------|
|      | Batch size (Per-GPU)           | 48          | 24        | 24    | 32        |
|      | Samples per iteration (Global) | 336         | 168       | 168   | 224       |
| DDPO | Gradient updates per iteration | 2           | 2         | 2     | 4         |
|      | Clip range                     | 1e-5        | 5e-5      | 1e-4  | 1e-4      |
|      | Optimizer Learning Rate        | 6e-4        | 6e-4      | 3e-4  | 3e-4      |

**Animal Prompts** dataset consists of 398 animal labels extracted from *ImageNet-1k class labels* Deng et al. (2009), often with comma-separated synonyms and scientific names.

**Comprehensive Prompts** dataset consists of 300 detailed and diverse descriptions of animals, vehicles, pieces of furniture, and landscapes with designated backgrounds, dynamics, or textiles.

### F.2  MIXGRPO EXPERIMENTS

We follow the setup of Li et al. (2025), letting reward model be "multi_reward" with equal weights. We set $T = 15$ as the denoising steps, AdamW optimizer with learning rate 1e-5 and weight decay 1e-4. For GRPO, the generation group size is 12 and clip range is 1e-4. We perform 12 gradient updates per iteration.

## G  LLM USAGE

Large Language Model (LLM) assists in LaTeX graphic alignments, spelling checks, and solving environment conflict issues in implementing DDPO and MixGRPO.

# H MORE EXPERIMENT RESULTS

## H.1 DDPO IMAGES ON DIFFERENT TRAINING STEPS

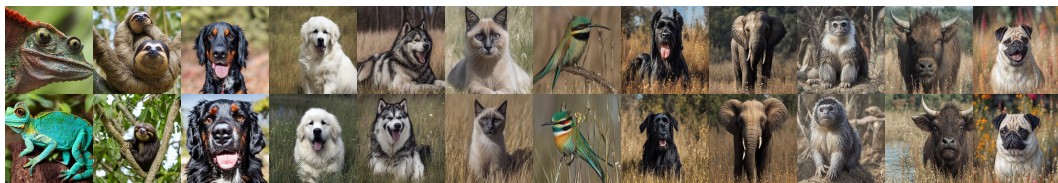

**Figure 7:** SDE (top) and ODE (bottom) sampling from every 100 training steps under PickScore with $\eta = 1.2$. Prompts (from left to right): "African chameleon, Chamae", "Gordon setter", "Great Pyrenees", "malamute, malemute, Alask, Bra", "Siamese cat, Siamese", "bee eater", "gaint schnauzer", "Indian elephant, Elephas", "marmoset", "water buffalo, water ox", "pug, pug dog".

## H.2 DDPO IMAGES UNDER IMAGEREWARD

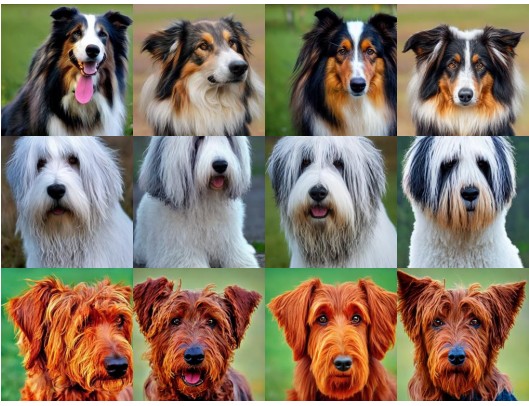

**Figure 8:** Sampling schemes (from left to right): (i) SDE with $\eta = 0.75$, (ii) ODE with $\eta = 0.75$; (iii) SDE with $\eta = 1.2$, (iv) ODE with $\eta = 1.2$.
Prompts (from top to bottom): "collie", "old English sheepdog, bob", "Irish terrier".

## H.3 DDPO IMAGES UNDER PICKSCORE

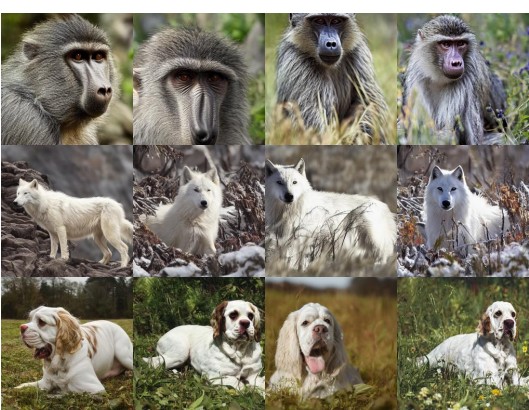

**Figure 9:** Sampling schemes (from left to right): (i) SDE with $\eta = 0.75$, (ii) ODE with $\eta = 0.75$; (iii) SDE with $\eta = 1.5$, (iv) ODE with $\eta = 1.5$.
Prompts (from top to bottom): "baboon", "white wolf, Arctic wolf", "clumber, clumber spaniel".

## H.4 DDPO IMAGES UNDER MORE COMPREHENSIVE PROMPTS

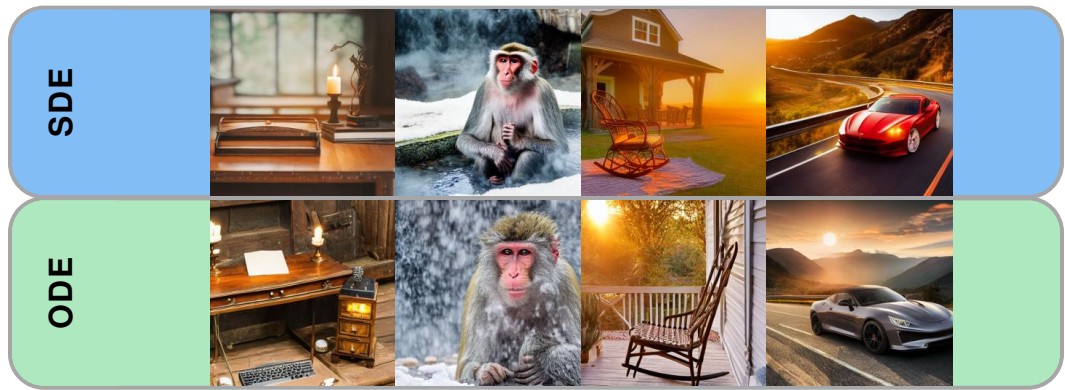

**Figure 10:** ODE (below) image generation preserves prompt instructions with better quality on details compared to SDE (above) image generation under large stochasticity ($\eta = 1.2$).

Prompts (from left to right): "A vintage writing desk with an open journal and a flickering candle.", "A macaque soaking in a steaming hot spring, surrounded by falling snow.", "A wicker rocking chair on a wrap-around porch during golden hour.", "A sleek sports car drifting on a mountain highway during golden hour."

## H.5 DDPO REWARD GAPS FOR OTHER REWARDS

### H.5.1 IMAGEREWARD

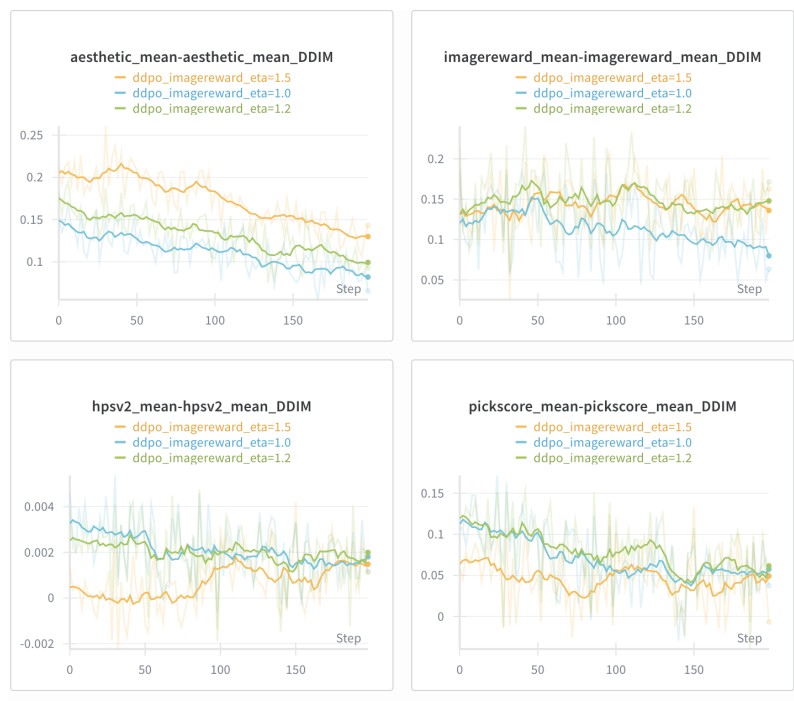

**Figure 11:** Bounded reward gaps trained under ImageReward for 200 steps with stochasticity $\eta \in \{1.0, 1.2, 1.5\}$

### H.5.2 HPSv2

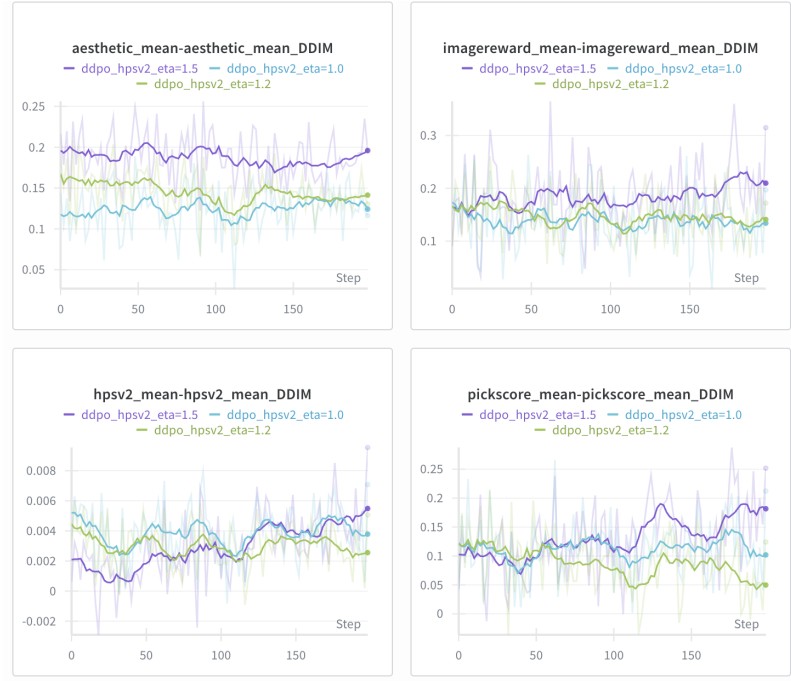

**Figure 12:** Bounded reward gaps trained under HPSv2 for 200 steps with stochasticity $\eta \in \{1.0, 1.2, 1.5\}$

### H.5.3 AESTHETIC

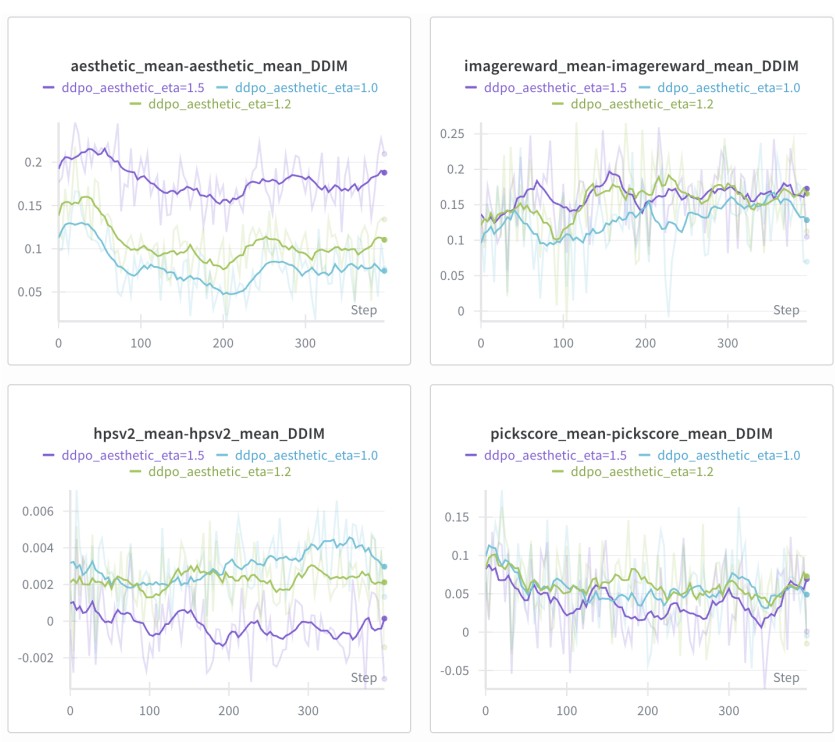

**Figure 13:** Bounded reward gaps trained under Aesthetic for 200 steps with stochasticity $\eta \in \{1.0, 1.2, 1.5\}$

