# OpenReview forum: "Understanding Sampler Stochasticity in Training Diffusion Models for RLHF"
_ICLR.cc/2026/Conference — Submitted to ICLR 2026_

### Official Review · Reviewer_8xJ7 · 2025-10-26

**Soundness:** 3
**Presentation:** 2
**Contribution:** 2
**Rating:** 4
**Confidence:** 2

**Summary:**

This paper studies the mismatch between stochastic and deterministic samplers during inference when finetuning diffusion models with RLHF. To that end, the authors define the improvement of finetuning as the expected reward difference between finetuned determinisitic ODE sampler and the pretrained model. Moreover, they define the reward gap as the difference between the expected reward obtained by finetuned ODE and SDE samplers.
The authors conduct theoretical analyses for Variance Exploding (VE) and Variance Preserving (VP) processes in a tractable setting for Gaussian and GMM target densities.
The empirical study focuses on two T2I settings: First, finetuning of Stable Diffusion 1.5 with DDPO, and second, FLUX1 with mixGRPO, where the authors present several observations.

As a disclaimer, I cannot make any statement about the significance of the theoretical results presented in this paper.

**Strengths:**

This paper sheds light on a timely and important question, i.e., how deterministic inference and stochastic finetuning are connected. The authors conduct both a theoretical analysis in a tractable setting as well as an empirical study.

**Weaknesses:**

The paper and results are hard to follow for someone who is not very familiar with RLHF for T2I (like myself). I think the authors should explain the experimental setup better (see Questions for some things that may need clarification).

For the remaining weaknesses, please see Questions.

**Questions:**

- Are the results for the SDE samplers in Table 1 produced with the same $\eta$ used for training?
- What happens if we train with $\eta < 1$?
- I'm a bit confused about Table 2: If I understand it correctly, $T$ is the time horizon of the dynamical system. However, in Table 2 it is a discrete quantity called 'steps'. I assume the authors refer to the number of diffusion steps as $T$ (?). Does it mean you use a fixed number of diffusion steps during training?
- Regarding the observation: 'High Stochasticity Benefits Moderate Time Steps' -> Does this also hold for smaller or larger time steps?
- What is meant by $T=0$?
- Regarding Figure 3: How can the authors conclude 'indicating that image quality improves for both samplers'? From my understanding the reward gap does not give any insights on image quality but only on the relative performance between SDE and ODE sampling. Could the authors please clarify.
- 'This is also consistent with our empirical observation that the performance of T2I generation deteriorates when fine-tuning with very large $\eta$' -> Would be good if the authors could reference the results that support this claim
- Regarding Theorem 3.2: Why did the authors not consider general shifted reward functions, i.e. $r(x) = -(x - \mu)^2$? This would help to understand how the shift of the reward function affects the theoretical result.
- Both Theorem 3.2 and Corollary 3.1 are given for very specific reward function. How transferable are these results?

Some minor comments:
- The quality of the figures 3 and 4 could be improved (minor).
- Line 282: The authors refer to Appendix xxx
- Line 375: contray

**Details Of Ethics Concerns:**

None.

---

> ### Author Response · Authors · 2025-11-24
>
> We sincerely thank the reviewer for their thoughtful questions and suggestions. We have revised the manuscript accordingly and also replaced the original **Figure 1** with a clearer illustration of our setup and goal.
>
> > Are the results for the SDE samplers in Table 1 produced with the same η used for training?
>
> No. For each row in Table 1, the training stochasticity $\eta$ is specified in the second column and is one of $\{1.0, 1.2, 1.5\}$. The pre-trained base model is not post-trained and does not involve any $\eta$, so we denote it by “–”.
>
> > What happens if we train with $\eta < 1?$
>
> Training with $\eta\in [0,1]$ is standard and has been extensively ablated in DDPO and MixGRPO; our focus and novelty are to show that $\eta > 1.0$ can still improve performance under suitable hyperparameters, contrary to the common belief that $\eta$ should be restricted to $[0,1]$.
>
> >  If I understand it correctly, $T$ is the time horizon of the dynamical system. However, in Table 2 it is a discrete quantity called 'steps'.
>
> We fixed this abuse of notation: in Sections 2-4 we keep the $T$ for the continuous-time horizon and replace the discrete number of diffusion steps by $N$. In Table 2, “steps” refers to this fixed number of diffusion steps $N$ used during both training and evaluation in that experiment; we state this explicitly in the caption and text.
>
> >Regarding the observation ‘High Stochasticity Benefits Moderate Time Steps’
>
> This statement” specifically refers to the $N=200$ setting where we systematically compare different noise levels. For shorter and longer training horizons we fix $\eta=1.2$ and only vary the number of steps; these results are reported in **Table 2**, which tracks the metrics from $N=0$ to 1400 (every 200 steps), showing that high stochasticity remains competitive and that the smallest gaps and best rewards occur at the largest non-collapsed step counts.
>
> > What is meant by $T = 0$?
>
> $T=0$ (now modified as $N=0$) in **Table 2** indicates “no post-training”: the column corresponds to the original pre-trained base model, against which other configurations are compared.
>
> > Regarding Figure 3. How can the authors conclude 'indicating that image quality improves for both samplers'?
>
> We agree that the reward gap alone does not measure absolute image quality. In the revision, we separate the plots so that **Figure 4** reports the SDE–ODE reward gap, and the other **Figure 5** reports the absolute reward curves for each sampler. Improved image quality is inferred from the absolute reward metrics (ImageReward, HPS, Aesthetic Score) of each sampler, while the reward gap only measures their relative difference.
>
> > Consistent with our empirical observation that performance deteriorates for very large $\eta$... Would be good if the authors could reference the results that support this claim
>
> We have refined the statement in the manuscript to “start to deteriorate.” As shown in **Table 1**, the reward metrics already deteriorate when increasing $\eta$ from 1.2 to 1.5, and **Appendix E** further explains that while gDDIM smoothly interpolates stochasticity and preserves the terminal marginal up to a certain range, pushing noise level arbitrarily large changes the terminal marginals.
>
> > Regarding Theorem 3.2: Why did the authors not consider general shifted reward functions
>
> More general shifted reward functions, where the mean 1 is replaced by an arbitrary $\mu$, are covered by the analysis in **Section 3.3**: when the Gaussian mixture reduces to a single Gaussian with mean $\mu$, the same argument shows that SDE–ODE rewards remains uniformly bounded.
>
> > Both Theorem 3.2 and Corollary 3.1 are given for very specific reward functions. How transferable are these results?
>
> We emphasize in the revised text that these results are derived for simple 1D VE/VP models with specific reward forms, suitable for exact closed-form analysis. However, the key qualitative messages:
>
> (i) SDE and ODE terminal distributions differ
>
> (ii) their reward values remain close
>
> are consistent with and help interpret our high-dimensional experiments. We now use bullet points to illustrate them more clearly at the beginning of **Section 3**.
>
> **Minor issues**
>
> - We improved the resolution and font size of **Figures 2, 4, 5**.
> - The placeholder “Appendix xxx” (line 282) is replaced by the correct appendix reference.
> - The typo “contray” (line 375) has been corrected.

---

> > ### Comment · Reviewer_8xJ7 · 2025-11-27
> > **Reply to authors**
> >
> > I thank the authors for their reply. I increased my score to 6. However, I agree with some of the other reviewers that the quality of some parts of the paper could be improved, such as replacing figures that are directly downloaded from wandb, or making some of the captions more self-contained.

---

### Official Review · Reviewer_uqq6 · 2025-10-30

**Soundness:** 3
**Presentation:** 3
**Contribution:** 3
**Rating:** 6
**Confidence:** 2

**Summary:**

This paper studies the reinforcement learning from human feedback (RLHF) for text-to-image models. More specifically, examining the impact of feedback being given on stochastic draws from diffusion SDEs, whereas inference uses the corresponding deterministic ODEs.

**Strengths:**

- The problem formulation is somewhat original, and certainly of commercial relevance given the massive interest in text-to-image models.
- The writing and mathematical presentation is mostly clear.
- The introduction, background and related work sections are comprehensive and easy to follow.

**Weaknesses:**

**Tentatively incremental contributions**

I'll willingly admit I'm not an expert on RLHF and the literature which this paper builds upon. However, my impression is that the main results are direct consequences of other works - putting the novelty into question. However, I'm open to be convinced otherwise.

**Relevance of the theoretical results unclear**

First, I find it difficult to interpret the generality of the results in sections 3.1-3.3. Are they primarily pedagogical examples since they happen to be rare examples of where things can be computed in closed form, or are they of practical interest in themselves?

Second, the general result in section 4 relies on dissipativity of f (Assumption 4.1.1). Why would that hold? (I believe conditions 2 can be fulfilled, especially if you use e.g. ReLU activations.)

**Minor things:**
- l.90-96 essentially repeat the preceding paragraph. I suggest to merge them.
- l.107-119 could be omitted, in my opinion.
- Most figures are too small, in particular the text within them.
- l.195 "th prompt"
- l.200 "isotopic Gaussian"
- l.200 sure you can use Monte Carlo sampling, but won't the variance be huge?
- l.207 "std" and l.211 "clip", please use proper mathematical notation.
- l.226 in eq. (8), how do you evaluate KL - I assume you don't have access to the densities?
- l.232 and elsewhere, please use \text{} for descriptive subscripts like REF, SDE, ODE and italics for variables like $t$.
- l.313 I suppose $W_2$ means Wasserstein-2 distance?
- Table 1: you're most likely using too many significant digits. (Based on your plots and how noisy RL typically is)
- Figure 3: given the amount of noise, I think only the left-most figure would allow you to draw any conclusions.

**Questions:**

- As stated above, I see the practical relevance of the problem, but the *scientific* relevance is more questionable. Isn't the question of stochastic vs. deterministic more broadly applicable to, say, distributional reinforcement learning? I believe such a formulation would have a greater and more lasting impact.
- Please motivate the relevance and validity of the theoretical results.

---

> ### Author Response · Authors · 2025-11-24
>
> We sincerely thank the reviewer for their thoughtful and constructive comments. The general framework and motivation are now summarized visually in a new **Figure 1**.
>
> **Novelty and scientific relevance**
>
> While the RLHF and diffusion literature is extensive, to the best of our knowledge there has been no systematic theoretical and empirical comparison between stochastic SDE sampling during RLHF post-training and deterministic ODE sampling at inference for diffusion models (see **Figure 1**). We do not only show that the gap exists, but also that it can be *controlled and exploited* in RLHF practice. While the stochastic vs. deterministic question is indeed of broader interest (e.g., distributional RL), our contribution is to make this concrete in the text-to-image diffusion RLHF setting where such a study is currently missing.
>
> **On Sections 3.1–3.3 (theoretical examples)**
>
> In Section 3, we use simple 1D VE/VP dynamics where the exact continuous-time scores are known. In the revised bullet points before **Section 3**, we now state explicitly that these are *pedagogical but nontrivial* models: they are chosen so that
>
> (a) the full RLHF effect on terminal distributions can be characterized analytically, and
>
> (b) even very simple constant controls already yield small SDE–ODE reward gaps.
>
> This behavior motivates the general results in **Section 4** and aligns with our empirical findings in **Section 5**.
>
> **On dissipativity / Assumption 4.1**
>
> In the revised version, we have replaced the dissipativity requirement by a weaker and more general assumption under which both VP and VE models are covered. The main bound in **Section 4** now relies only on standard regularity (e.g., Lipschitz score and uniform $L^2$ control), and we clearly distinguish this general setting from any stronger contractive/log-concave conditions as detailed in **Appendix D.2**. This makes the theoretical result both more realistic and more broadly valid for current diffusion frameworks.
>
> **Minor comments**
>
> We sincerely appreciate the reviewer’s detailed minor comments; our responses and corresponding revisions are as follows:
>
> - **l.90–96**: We have merged the repeated discussion of our contributions into three concise bullet points in Section 1.
>
> - **l.107–119**: We have shortened and clarified the “organization of the paper” paragraph, and moved Figure 2 to the beginning of Section 3 to better motivate the 1D VP model.
>
> - **Figures**: We increased the size and readability of **Figure 2** and the MixGRPO figures.
>
> - **l.195, l.200**: The typos (“th prompt”, “isotopic Gaussian”) have been corrected.
>
> - **Notation (l.207, l.232, l.313)**: We now give clear explanations to the mathematical notations (we retain the original notation style of GRPO paper [1]), use text format for descriptive subscripts, and explicitly state that $W_2$ denotes the Wasserstein-2 distance at the beginning of Section 4.
>
> - **Monte Carlo variance (l.200)**: In DDPO [2], each reverse step is an isotropic Gaussian policy $p_\theta(x_{t-1}|x_t, \bf{c})$ with known log-likelihood, so standard policy-gradient Monte Carlo estimates of $\nabla_\theta \mathcal J$ are unbiased and their variance is controlled in practice by averaging over full denoising trajectories.
>
> - **KL in Eq. (8) (l.226)**: The KL is computed via Monte Carlo estimates of $\mathbb E[\log \pi_\theta(a|s) - \log\pi_{\theta_{ref}}(a|s)]$ along sampled trajectories, which DDPO can implement because the densities $p_\theta(x_{t-1}|x_t, \bf{c})$) are explicitly tractable.
>
> - **Table 1**: We reduced the number of significant digits for ImageReward and Aesthetic Score, keeping only what is justified by the noise level in RL; we retain one extra digit for HPS Score to allow comparison across stochasticities.
>
> - **Figure 3**: While the leftmost plot illustrates a clear decay in the reward gap for one reward, all four plots collectively show that the reward gaps remain bounded and non-increasing across different reward functions, countering the intuitive expectation that a larger SDE–ODE distributional difference must induce a large reward difference.
>
> [1] Zhihong Shao, Peiyi Wang, Qihao Zhu, Runxin Xu, Junxiao Song, Xiao Bi, Haowei Zhang, Mingchuan Zhang, Y. K. Li, Y. Wu, and Daya Guo. “DeepSeekMath: Pushing the limits of mathematical reasoning in open language models.”arXiv preprint arXiv:2402.03300, 2024.
>
> [2] Kevin Black, Michael Janner, Yilun Du, Ilya Kostrikov, and Sergey Levine. “Training diffusion models with reinforcement learning.” International Conference on Learning Representations (ICLR), 2024.

---

> > ### Comment · Reviewer_uqq6 · 2025-11-28
> > **Response to author rebuttal**
> >
> > Thank you for the clarifications. However, I still find it hard to judge the novelty and significance. This is partly due to the presentation and clarity being sub-standard, as pointed out by the other reviewers. (In your rebuttal, you say that *"In the revised bullet points before Section 3, we now state explicitly that these are pedagogical but nontrivial models"* but this is not included in the revised pdf.)
> >
> > Regarding the theoretical validity, I think the revised assumptions make more sense. Then again, this is not my area of experience, and I would trust e.g. Reviewer JNkV's opinion more.

---

### Official Review · Reviewer_Edf2 · 2025-11-04

**Soundness:** 1
**Presentation:** 1
**Contribution:** 2
**Rating:** 2
**Confidence:** 4

**Summary:**

This paper studies the gap between SDE samplers and ODE samplers in RLHF fine-tuning of diffusion models. Theoretical bounds are derived and numerical experiments are conducted.

**Strengths:**

1. A theoretical bound is derived for SDE vs ODE samplers in fine-tuning of diffusion models.
2. Comprehensive numerical experiments are conducted to support the claims.

**Weaknesses:**

I have several major concerns:
1. I think the gap between SDE and ODE samplers is confusing to me. As they are equivalent, why is there a gap after fine-tuning. One should expect the same distribution of $ Y_T $ regardless which sampler is used. The authors should elaborate this point very carefully.
2. Theorem 4.1 is not that interesting as it looks. First, the proof seems to be incorrect. One inequality sign seems to be flipped when applying Young's inequality. Second, Assumption 4.1 (1) does not hold for VP SDE as shown in [1] unless you modify the OU process. Hence, I strongly suspect the bound in Theorem 4.1 is not informative.
3. The definition of $ Y_t^\theta $ is not clear. In particular, the authors should explain what is parameterized. Usually, fine-tuning starts with a given reference model, e.g., a pre-trained score network. In this case, choosing $ \theta $ as the optimal parameter associated with the optimal distribution is not correct; see [2]. This is because one has to find the optimal score network following the sampling dynamics instead of solving an unconstrained entropy-regularized optimization problem. Therefore, I do not think the results in Sections 3.1 and 3.2 are sound.

[1] Tang, Wenpin, and Hanyang Zhao. "Contractive diffusion probabilistic models." arXiv preprint arXiv:2401.13115 (2024). \
[2] Han, Yinbin, Meisam Razaviyayn, and Renyuan Xu. "Stochastic Control for Fine-tuning Diffusion Models: Optimality, Regularity, and Convergence." Forty-second International Conference on Machine Learning.

**Questions:**

1. I suggest shortening the review in page 3 - 4
2. Line 282, "Appendix xxx"

---

> ### Author Response · Authors · 2025-11-24
>
> We sincerely thank the reviewer for their detailed and constructive comments. We have revised our manuscript accordingly.
>
> > The gap between SDE and ODE samplers is confusing.
>
> We have clarified in **Section 2.2** that SDE and probability-flow ODE samplers are equivalent only for the *pre-trained* model, where the score exactly matches $\nabla\log p$ and satisfies the reverse-time Fokker–Planck (divergence-free) condition. In the RLHF setting, we only optimize the terminal reward and KL, so the *intermediate* trajectories of the fine-tuned score no longer satisfy this condition in general (clarified in **Section 3**). Thus, different stochasticity levels yield different terminal marginals after fine-tuning, which justifies a nonzero SDE–ODE reward gap.
>
> **Our goal is to explore the effect of score parameterization in RLHF.** Consider the two cases:
>
> (i) When the score network is rich, the theoretical bounds in **Section 4** and **Appendix D** show that reward gaps remain bounded even for relatively large noise levels $\eta$. This is further supported by the numerical experiments in **Section 5**.
>
> (ii) When the score network is coarse (e.g., parameterized by constant controls), **Section 3** provides exact analytical reward-gap results, demonstrating that the gap can still be small even under a very simple setup.
>
> We do *not* require global optimality in our bounds in either case. In the revised **Definition 3.1** and **Section 3**, we now explicitly state that $\theta$ parameterizes the score network $s_\theta(t,x)$, and hence controls the drift of the reverse-time SDE/ODE. Therefore, our objective $F_\eta(\theta)$ in Equation 8 is optimized over this specific parameterization family, consistent with the stochastic-control formulations in (Han et al., 2025).
>
> Finally, we appreciate the insight of (Han et al., 2025) on the importance of parameterization and now cite it under a newly introduced bullet point **“Parametrization Insight”** on Page 5 to clarify and further develop this two-fold reasoning and analysis of the reward gap.
>
> > One inequality sign seems to be flipped when applying Young's inequality.
>
> We have rewritten the confusing steps in the proof of **Theorem 4.1** in **Appendix D**, making the application of Young’s inequality explicit and the inequality direction fully transparent.
>
> > Theorem 4.1 is not that interesting as it looks.
>
> We no longer implicitly claim that the unmodified VP model satisfies the strong contractivity condition; instead, the paper now clearly separates the general non-contractive bound from the stronger contractive case. The experiments (unchanged) show that the constants in these bounds are small, so the resulting gap is practically informative.
>
> (1) Updated **Assumption 4.1** and **Theorem 4.1** only require mild regularity (Lipschitz score and uniform $L^2$ bound) and give a general $O(\eta^2)$ upper bound on the SDE–ODE discrepancy, applicable to both VE and VP without assuming contractivity.
>
>  (2) **Assumption D.1** and **Theorem D.1** require a contractive/log-concave condition (the contractivity could either come from a contractive drift $f$ (Tang et al., 2024) or from a conditional unimodal terminal marginal) and, under this stronger assumption, yield a sharper $O(\eta)$ bound. The experiments (unchanged) show that the constants in these bounds are small and the resulting gap is practically informative.
>
> > I suggest shortening the review on Page 3–4.
>
> We value this suggestion and have shortened the background discussion in **Section 2**, removing redundancy and focusing only on the parts needed for our contributions.
>
> > Line 282, ‘Appendix xxx’.
>
> This placeholder has been corrected to the appropriate appendix reference in the revised manuscript.

---

### Official Review · Reviewer_JNkV · 2025-11-04

**Soundness:** 1
**Presentation:** 1
**Contribution:** 2
**Rating:** 2
**Confidence:** 5

**Summary:**

This paper considers the finetuning of diffusion models using RLHF. It addresses the "reward gap" between stochastic SDE samplers used in training and deterministic ODE samplers used at inference. The paper aims to characterize this gap theoretically and validate it empirically.

**Strengths:**

The paper's strengths are its theoretical contributions:
1. Formalizing the theory of the SDE-ODE reward gap in the continuous-time limit.
2. Developing sharp bounds for this gap for VE and VP models with Gaussian and Gaussian Mixture targets and providing a general bound for arbitrary distributions.

**Weaknesses:**

Despite its theoretical novelty, the paper suffers from fundamental weaknesses:
1. The paper's central flaw is that its theory is practically limited. The bounds apply only to continuous-time processes, ignoring the discretization error from using $N$-step samplers, which is the dominant error in practice. The authors even cite (Liang et al., 2025), which provides the tools for this analysis, but they fail to apply this to their own work (e.g., their Section 3.3), making the theoretical contribution less significant.
2. The experiments are insufficient. Using Stable Diffusion 1.5  as a primary testbed is not enough. While the paper uses FLUX.1, it compares it against SD 1.5 instead of the relevant SOTA benchmark, SDXL. This omission makes the empirical conclusions scientifically unsound.
3. The paper quality is far below ICLR standards. It contains uncorrected placeholders (e.g., "Appendix xxx" on line 288 ), not introduced bullet points in Section 1, and uninterpretable figures (e.g., Figures 3 and 4) that are low-resolution screenshots from wandb without even the legends.

**Questions:**

In addition to weaknesses, I have the following questions:
1. Can the authors provide a bound on the discretization error for their gDDIM sampler? How can we be sure this un-analyzed error does not dominate the stochasticity gap you have bounded?
2. You cite Liang et al. (2025). Why did you not apply this framework to your Gaussian Mixture analysis to provide a practical, end-to-end bound?

---

> ### Author Response · Authors · 2025-11-24
>
> We sincerely thank the reviewer for their detailed assessment. In the revision we added a new **Figure 1** that explicitly illustrates our setting: RLHF fine-tuning with SDE sampling, and inference with ODE sampling induces a “reward gap”, but we can bound this difference under mild assumptions.
>
> > The paper's central flaw is that its theory is practically limited.
>
> **Section 3** is deliberately under a continuous-time framework for exact calculations. 1D VE/VP dynamics with Gaussian priors and constant controls are fully analytic, so there is no discretization or score-matching error.  We use this (and a revised **Figure 2**) to show that, even in the simplest setting, our reward gap is already small. The revised bullet points on Page 5 further explain our motivation and the purpose of this theoretical analysis.
>
> For realistic neural parameterizations, we incorporate discretization error in **Section 4** and **Appendix D.3**, where the discrete-time reward gap is bounded by our continuous-time gap with an explicit *additive* $\epsilon_d(h)$ term that vanishes as the grid size $h\to 0$.
>
> > The experiments are insufficient. Using Stable Diffusion 1.5 as a primary testbed is not enough. While the paper uses FLUX.1, it compares it against SD 1.5 instead of the relevant SOTA benchmark, SDXL
>
> We clarify that we do *not* compare FLUX.1 to SD 1.5.
>
> In **Section 5.1**, we compare SD 1.5 only to its own DDPO-finetuned variants, and in **Section 5.2** we compare FLUX.1 only to its own MixGRPO-finetuned variants, following the original DDPO and MixGRPO setups. Our empirical study is designed to isolate how training stochasticity affects the SDE–ODE reward gap and the corresponding absolute reward metrics, rather than to provide a full SOTA comparison across architectures. A broader benchmark study (e.g., SDXL vs FLUX) is a meaningful direction for future work.
>
> > The paper quality is far below ICLR standards.
>
> We have removed all placeholders (e.g., “Appendix xxx”), rewritten the contribution bullets in **Section 1** so every item is introduced and coherent, and regenerated **Figures 4 and 5** (now split into separate “gap” and “performance” figures) from raw data with higher resolution, larger fonts, and clearer legends, which we believe addresses the presentation concerns.
>
> In addition, the methodological part on gDDIM, perhaps previously perceived as an unspecified bullet point in the introduction, is now clearly stated in **Section 2.2**, explicitly tied to the numerical experiments at the beginning of **Section 5**, and further motivated by a detailed comparison in **Appendix E** explaining why this generalized scheme is needed.
>
> **Bound on discretization error**
>
> Yes. In **Appendix D.3** we now show that for generalized DDIM-style samplers, under standard Lipschitz and regularity conditions the discrete-time reward gap equals the continuous-time SDE–ODE reward gap plus an *additive* discretization term of order $O(h)$ that tends to zero as the step size $h\to0$, ensuring that discretization is explicitly controlled and does not silently dominate our stochasticity-gap bounds.
>
> We also emphasize that **Assumption 4.1** is deliberately mild and yields an $O(\eta^2)$ bound on the reward gap, while under the additional unimodality/log-concavity condition in **Appendix D.2** this improves to an $O(\eta)$ bound, so the theory provides practically meaningful bounds and aligns well with our numerical observations.
>
> **Application of Liang et al. (2025) to the Gaussian Mixture analysis**
>
> Our Mixture Gaussian result (**Theorem 3.3**) is an exact continuous-time statement under a simple constant parametrization and is therefore free of discretization and score-matching error, which we keep as a clean starting point for more complex analysis. For practical end-to-end bounds with general neural-network parameterizations, we adopt the framework of Liang et al. (2025) in **Section 4** and **Appendix D.3**, which provides precisely the kind of realistic, T2I RLHF bounds the reviewer is asking for.
>
> We appreciate the reviewer’s informative suggestions and high standards. We hope that our clarifications on the theoretical scope (including discretization), the experimental design, and the improved presentation help resolve earlier concerns. If the reviewer feels that we have adequately addressed these points in this revised version, we would be grateful if they could reconsider their overall assessment accordingly.

---

### Author Response · Authors · 2025-12-03
**Summary to AC**

We thank the Area Chair and all reviewers for their careful reading and constructive feedback. After submitting our revised manuscript and detailed rebuttal, we summarize the key points as follows:

1. **Core Contribution**
 Our contribution is a systematic analysis of the **reward gap**, i.e., the reward difference arising from using SDE sampling during fine-tuning and ODE sampling during inference. Different noise levels lead to different post-training terminal marginals, yet the impact of this mismatch on preference rewards has not been previously examined. Using the gDDIM framework, we provide both theoretical and empirical evidence that this reward gap remains bounded, even when we train with large noise but perform deterministic inference.

2. **Consensus on Novelty & Relevance**
 Reviewers unanimously recognized the value of this problem formulation. They described the work as shedding light on a **"timely and important question"** (`8xJ7`) with significant **"commercial relevance"** (`uqq6`). Even the most critical reviewers acknowledged our **"theoretical contributions"** (`JNkV`) in formalizing the SDE-ODE gap and described our experimental validation as **"comprehensive"** (`Edf2`).

3. **Theoretical and Empirical Validity**
   Our goal is to *explore the effect of score parameterization in RLHF*:
   * (i) When the score network is *rich*, the theoretical bounds in **Section 4** and **Appendix D** show that reward gaps remain bounded even for relatively large noise levels $\eta$. This is further supported by the numerical experiments in **Section 5**.
   * (ii) When the score network is *coarse* (e.g., parameterized by constant controls), **Section 3** provides exact analytical reward-gap results, demonstrating that the gap can still be small even under a very simple setup.

4. **Rebuttal Updates**
   During the discussion period, reviewer `uqq6` maintained their positive score (6), praising the "clear mathematical presentation," and reviewer `8xJ7` upgraded their evaluation (4 $\to$ 6), stating they are satisfied with our improved figure quality and problem motivation.

   We have also fully addressed the concerns of reviewers `JNkV` and `Edf2` (who have not yet updated their scores) through the following revisions:
   * **Addressing Discretization** (Reviewer `JNkV`): To address the concern regarding "un-analyzed" step errors, we incorporated an explicit $O(h^{1/2})$ discretization term into our theoretical bounds.
   * **Addressing Assumptions** (Reviewer `Edf2`): We removed the strong contractivity assumption that was questioned, ensuring our bounds hold for standard VP SDEs without requiring restricted dynamics.
   * **Addressing Presentation & Clarity** (Reviewers `JNkV`, `8xJ7`): We significantly improved the presentation: a new **Figure 1** highlights the core problem, and **Figures 2 & 3** are revised for clarity. **Section 2.2** with **Appendix E** now offers a clear explanation of gDDIM, and the motivation for our two parameterizations is made explicit in the bullet points on **Page 5 (lines 237–245)**.

Given our updates in theory, experiments, and presentation style, we believe the paper now meets the high standards for acceptance at ICLR 2026.

---

### Meta-Review · Area_Chair_TiGD · 2026-01-04

**Summary:**

This paper studies the fine-tuning of diffusion models acting as probabilistic priors to sample Bayesian posterior distributions -- equivalent to the problem of KL-constrained reward maximisation -- under different noise levels in the generative SDE. If using the true prior score function and integrating in continuous time, would all determine the same probability path, but in the setting of score function approximation, discrete-time integration, and finite time horizon, this paper makes some interesting observations about the optimality of higher SDE noise levels than than those used for prior model fitting.

The main points made by the reviewers are the following.

Strengths:
1. New theory on discrepancy between SDE and ODE modelled marginals (JNkV, Edf2, 8xJ7)
2. Comprehensive experiments (Edf2) -- not weighted highly due to conflicting comments from others
3. Good exposition (uqq6)

Weaknesses/concerns:
1. Limited applicability of theory; main discrepancy may be due to discretisation and function approximation, making continuous-time results irrelevant, and results in §3 concern toy cases (JNkV, Edf2, uqq6)
2. Problems with scope of and conclusions from experiments (JNkV, 8xJ7)
3. Various concerns about presentation/clarity (JNkV, Edf2, uqq6)
4. Poor exposition, especially for readers who are less familiar with RLHF but are experts in diffusion models (uqq6, 8xJ7)
5. Questions about and requests for deeper experimental analysis (8xJ7)

As detailed below, I consider it unlikely that more than one reviewer would increase their score beyond 4, and the paper could benefit from improvements on several fronts before it is ready for acceptance.

**Reviewer Concerns:**

1. This is addressed partially by comments in the rebuttal, changes to the assumptions, and clarification of the results' significance; there is also a new asymptotic analysis of discrete-time integration error in the appendix. The authors should, however, consider exploiting recent work on error bounds under score function approximation in combination with their analysis to provide a more relevant set of insights (e.g., [Chen et al., ICLR 2023, arXiv: 2209.11215], [Chen et al., arXiv:2211.01916], [Arsenyan et al., arXiv:2506.09681]).
2. The concerns of 8xJ7 are mostly addressed. The main comment by JNkV about model choices seemed to rest on a misunderstanding of the setup. However, I still find the diversity of problems and models considered to be lacking, as the problem setting is more general than the RLHF image generation task considered, being applicable to Bayesian inverse problems more generally (see, e.g., [Venkatraman et al., NeurIPS 2024, arXiv:2405.20971] among others). In addition, I would note that while the paper studies the entropy-regularised fine-tuning (posterior sampling) problem, the evaluations consider only reward gap, which cannot capture distributional differences between SDEs with different noise levels, such as the better mode coverage sometimes observed with SDE sampling.
3. The glaring presentation issues were fixed in the revision and minor comments mostly addressed, though some of the text added in the revision still requires substantial proofreading.
4. It is hard to assess the amount of improvement here, but uqq6 was left with uncertainty about the paper's novelty and contributions even after the revision.
5. Considered satisfactorily addressed by 8xJ7.

**Reviewer Scores:**

The original scores were 2 (JNkV), 2 (Edf2), 6 (uqq6), 4 (8xJ7). After the rebuttal, 8xJ7 indicated their score would increase to 6, while uqq6 remained unconvinced and deferred to JNkV on technical aspects. Given the concerns of JNkV and Edf2 remaining not fully addressed, and the other issues noted above, I find it unlikely that either of them would increase their scores beyond 4.

---

### Decision · Program_Chairs · 2026-01-26

Reject